# Locally synthesized glycyl aminoacyl-tRNA synthetase is important for local translation in neurons

Tyler Brent de Leon, Adi Golani-Armon, Bar Cohen, Yoav S Arava

**Regulation of gene expression is essential for neuronal development and function. A prominent regulatory mechanism involves synthesis of proteins at their activity site. Such local protein synthesis enables neurons to respond rapidly and tightly to stimuli. Key components of the translation machinery, including mRNA and ribosomes, were identified in subcellular regions of neurons. Yet, the role of tRNAs and their charging enzymes, aminoacyl-tRNA synthetases (ARS), in this process remains largely unclear. Here, we demonstrate that glycyl-tRNA synthetase (Gars1) mRNA is abundant in neurites and undergoes local translation, producing GARS1 protein. Notably, Gars1 mRNA colocalizes with mitochondria in a translation-dependent manner, with its coding sequence (CDS) sufficient to direct this association. The localized GARS1 protein is in close proximity to tRNA$^{Gly}$, and disrupting their proximity impairs local protein synthesis in neurites. These findings establish the functional importance of GARS1 and tRNA$^{Gly}$ in neuritic translation and highlight mitochondria as hubs for mRNA transport and translation.**

## Introduction

Precise spatiotemporal regulation of gene expression is essential for neuronal development and function (Martin & Ephrussi, 2009). This requirement is particularly pronounced in neurons because of their extreme polarity, with dendrites and axons extending far from the soma (Rangaraju et al, 2017). To achieve compartment-specific gene expression, subsets of mRNAs localize to distal neurites, where they undergo local translation in response to extracellular cues (Lin & Holt, 2007). Local protein synthesis provides neurons with the capacity to respond rapidly to stimuli and carry out specialized functions with high spatial precision (Cagnetta et al, 2023).

Advances in transcriptomics (Zivraj et al, 2010; Hafner et al, 2019; Perez et al, 2021) and translatomics (Shigeoka et al, 2016; Ouwenga et al, 2017; Biever et al, 2020) have revealed thousands of mRNAs localized to axons and dendrites, many of which are associated with ribosomes, indicating widespread local translation. High-resolution imaging further confirms the presence of translational machinery in axons, dendrites, and synapses (Cioni et al, 2019; Hafner et al, 2019; Sun et al, 2021; Koppers et al, 2024; Zhang et al, 2025). This localized translation supports diverse neuronal processes, including axon guidance (Martin, 2004; Swanger & Bassell, 2011), synapse formation and plasticity (Martin, 2004; Rangaraju et al, 2019), and regeneration after injury (Yoo et al, 2010; Terenzio et al, 2018). Disruption of local protein synthesis leads to defects in neuronal development (Preitner et al, 2014), synaptic plasticity (Kang & Schuman, 1996), and neuronal survival (Terenzio et al, 2018; Cagnetta et al, 2023). Moreover, dysregulation of this process has been implicated in neurological disorders such as autism spectrum disorders (Gkogkas et al, 2013), fragile X syndrome (Kao et al, 2010; Santini et al, 2017), and peripheral neuropathies (Fallini et al, 2016; Pease-Raissi et al, 2017).

Although mRNAs and translation factors have been extensively studied, the presence and role of tRNAs and aminoacyl-tRNA synthetases (ARSs) in neurites remain less understood. ARSs catalyze the charging of cognate tRNAs with amino acids, enabling translation (Rubio Gomez & Ibba, 2020). Some ARSs have been detected in distal neurites; for instance, glycyl-tRNA synthetase (GARS1) localizes to neurites in cultured neurons and peripheral axons in human tissue (Antonellis et al, 2006), whereas tyrosyl-tRNA synthetase (YARS1) is found in neurites of differentiating N2a cells and primary motor neurons (Jordanova et al, 2006). GARS1's functional significance is underscored by neuropathy-associated mutations that confer dominant-negative or gain-of-function properties, leading to peripheral nerve degeneration (Niehues et al, 2015; Sleigh et al, 2020; Vinogradova et al, 2021; Mora et al, 2025). However, whether these effects arise from disrupted local functions within neurites remains unresolved.

Here, we show that Gars1 mRNA localizes to both axons and dendrites of rat primary cortical neurons. The mRNA colocalizes with mitochondria in a translation-dependent manner, where its CDS is sufficient to direct this mitochondrial association. Gars1 mRNA is locally translated to the GARS1 protein and is expressed at comparable amounts to that of the highly localized neuritic protein, FMRP (Weiler et al, 1997; Antar et al, 2005). Furthermore,

Faculty of Biology, Technion – Israel Institute of Technology, Haifa, Israel

Correspondence: arava@technion.ac.il

GARS1 protein associates with tRNA[Gly] in neurites and disruption of this interaction by antisense oligonucleotides (ASOs) reduces neuritic protein synthesis. These findings highlight the critical role of GARS1–tRNA[Gly] association in supporting local translation within neurons.

# Results

### Gars1 mRNA is abundant in rat primary neurites

Although the GARS1 protein localizes to neurites in cultured human cells and neuronal tissues (Antonellis et al, 2006; Nangle et al, 2007), the subcellular distribution of Gars1 mRNA remains unexplored. To address this, we first analyzed existing transcriptomics datasets of axonal, dendritic, and synaptic mRNAs (Von Kügelgen & Chekulaeva, 2020). Gars1 mRNA was detected in axons of human iPSC-derived motor neurons (Maciel et al, 2018), primary spinal cord motor neurons (Briese et al, 2016; Rotem et al, 2017), dorsal root ganglia (Minis et al, 2014), Ascl1-induced neurons (Zappulo et al, 2017), and primary cortical neurons (Middleton et al, 2019). In all cases, we observed similar enrichment and abundance of Gars1 to Actb (Fig S1), a neurite-enriched mRNA (Bassell et al, 1998).

To directly visualize the localization of Gars1 mRNA, we performed single-molecule inexpensive fluorescence in situ hybridization (smiFISH) (Tsanov et al, 2016) using fluorophore-labeled probes in rat primary cortical neurons (Fig 1A and B). We compared the abundance of Gars1 mRNA in neurites with that of Actb mRNA and the soma-enriched Tubb3 mRNA (Zappulo et al, 2017). Confocal imaging revealed extensive Gars1 mRNA localization in neurites, comparable to Actb and significantly higher than Tubb3 (Fig 1A–C). The distribution of Gars1 and Actb (Bassell et al, 1998) mRNAs appears relatively constant along the neurites. Gars1 mRNA localized to both dendrites and axons (Fig S2). Consistent patterns were observed in differentiated N2a cells, where Gars1 mRNA levels matched those of Actb (Fig S3).

To further validate these observations, we used transwell assays to physically separate neurites from soma in differentiated CAD cells and performed RT–qPCR on extracted RNA (Arora et al, 2021; Cohen et al, 2022). Compared with the neurite-enriched Actb control, Gars1 mRNA showed comparable enrichment in neurite fractions (Fig 1E). Together, these results establish Gars1 mRNA as a highly abundant neuritic transcript across multiple neuronal models.

### Gars1 mRNA is locally translated in neurites

Having observed Gars1 mRNA in neurites, we next sought to determine the localization of the GARS1 protein. Immunofluorescence (IF) for the endogenous GARS1 protein in primary rat cortical neurons revealed clear localization to neurites, comparable to that of the highly localized fragile X mental retardation protein (FMRP) (Weiler et al, 1997; Antar et al, 2005) (Fig 2A–C). Likewise, we observed similar localization patterns in differentiated N2a cells (Fig S4A–C), consistent with previous observations (Nangle et al, 2007).

To investigate whether this protein is derived from the local translation of Gars1 mRNA, we tested the possible ribosomal association of the mRNA. For that, we combined smiFISH with IF for the ribosomal protein RPS3 in primary cortical neurons. Super-resolution imaging revealed extensive colocalization between Gars1 mRNA puncta and ribosomal clusters within neurites (Fig 2D and E). To associate this proximity with active ribosomal translation, we applied translation inhibitors, cycloheximide (CHX) or puromycin (Puro), before the smiFISH and IF analysis (Rodriguez et al, 2006). Cycloheximide treatment, which stabilizes the association of translating ribosomes with mRNAs (Schneider-Poetsch et al, 2010), resulted in a significant increase in colocalization between Gars1 mRNA and RPS3 (Fig 2D and E). Conversely, puromycin treatment, which causes dissociation of ribosomes from translating mRNAs (Blobel & Sabatini, 1971), decreased colocalization between Gars1 mRNA and RPS3 (Fig 2D and E). Similar colocalization trends between Gars1 mRNA, RPS3, and RPL10 were obtained in differentiated N2a cells (Fig S4D–G).

To directly detect newly synthesized GARS1 protein in neurites, we performed puromycilation followed by proximity ligation assay (Puro-PLA) (Tom Dieck et al, 2015) in primary rat cortical neurons. This technique allows visualization of nascent proteins by incorporating puromycin into newly synthesized peptide chains and codetecting it with antibodies recognizing GARS1 and puromycin (Fig S5A). In the absence of puromycin treatment, we observed little to no puromycin signal, validating the specificity of the antibodies that were used (Fig S5B). Upon puromycin treatment, we observed significant amounts of newly synthesized GARS1 protein in neurites, providing further evidence of local translation (Fig 2F and G). Ribo-seq data further showed that Gars1 mRNA associates with active ribosomes, corroborating our findings (Glock et al, 2021). Altogether, these findings establish that Gars1 mRNA is locally translated to GARS1 protein in neurites of primary rat neurons and in differentiated neuronal cell lines.

### Gars1 mRNA mitochondrial localization

The GARS1 protein localizes to both the cytoplasm and mitochondria (Alexandrova et al, 2015), consistent with its dual role in charging cytosolic and mitochondrial tRNA[Gly]. Although it was recently shown to associate with the mitochondrial outer membrane (OMM) in HEK293T cells (Qin et al, 2021), its association with mitochondria in neuronal neurites remains largely unexplored. To address this, we performed immunofluorescence (IF) staining for the GARS1 protein and assessed its spatial relationship with mitochondria. We observed a distinct overlap between GARS1 puncta and mitochondria in neurites (Fig 3A and B).

mRNAs encoding nuclear-encoded mitochondrial proteins are frequently associated with mitochondrial outer membrane, to facilitate their import (Cohen et al, 2024). We therefore investigated whether Gars1 mRNA associates with mitochondria in neurons. We first performed fractionation of crude mitochondria by differential centrifugation, followed by analysis of associated RNAs by RT–qPCR. Western analysis confirmed enrichment of mitochondrial fraction (ATP5A) and lack of a cytosolic marker protein (GAPDH) (Fig 3C). RT–qPCR analysis revealed that Gars1 mRNA is enriched in the mitochondrial fraction to levels comparable to

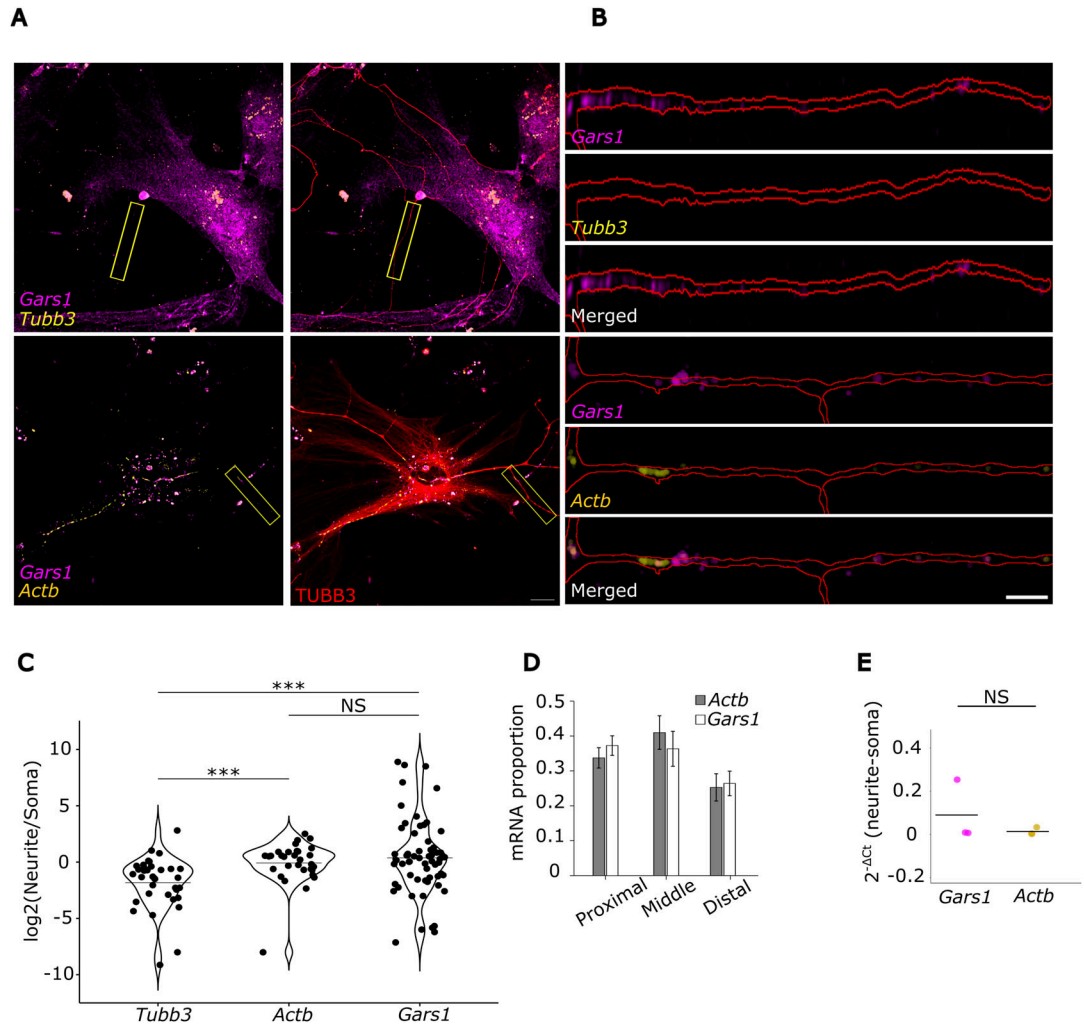

**Figure 1. *Gars1* mRNA localization in neurites.**
**(A)** Primary rat cortical neurons (5–7 DIV) were fixed and hybridized with probes against Gars1 (magenta), Actb (yellow), or Tubb3 (yellow) mRNAs and immunolabeled with α-TUBB3 (red) to visualize soma and neurites. Representative smiFISH images of Gars1, Actb, and Tubb3 mRNAs in neurites are shown. Scale bar, 20 μm. **(A, B)** Higher magnification view of the neurite in (A) showing mRNA distribution along its length. Neurite masks were generated from the TUBB3 signal. Scale bar, 5 μm.
**(C)** Quantification of mRNA density in soma and neurites using RS-FISH. Spot density was normalized to area, and neurite-to-soma ratios were calculated. Data are presented as individual points representing single neurites (*n* = 31–60 neurites, pooled from four independent biological replicates, ~3 cells per replicate).
**(D)** Proportional distribution of Tubb3, Actb, and Gars1 mRNA spot counts across neurite regions (proximal: first 20 μm from soma; distal: last 20 μm) (*n* = 31–60 neurites, pooled from four independent biological replicates, ~3 cells per replicate). **(E)** CAD cells grown on microporous membranes (30% confluency) were serum-starved (0.8% FBS) for 7–14 d to induce neurite differentiation. RNA was extracted from soma and neurite compartments and subjected to RT–qPCR. $\Delta C_t$ values were averaged from *n* = 3 independent biological replicates. **(C, D)** Data information: *P*-values: ***< 0.001; (NS) > 0.05 by the Wilcoxon rank-sum test (C); *t* test (D). **(C, D, E)** Results are presented as a violin plot (C) with median indicated by a horizontal line; mean ± SEM (D); mean (E).

Cox7c, an established mitochondrial protein mRNA (Cohen et al, 2022), relative to the cytosolic transcript Actb (Fig 3D).

To visualize this association in situ, we transfected cells with Mito-BFP to label mitochondria and performed smiFISH for endogenous Gars1 mRNA. Importantly, because the two Gars1 mRNA isoforms vary only by the short sequence encoding the signal peptide, our probe design cannot unambiguously differentiate between them. Consequently, the signals detected in these experiments represent the total pool of Gars1 transcripts. To minimize channel crosstalk during colocalization analysis we used a BFP-tagged mitochondria marker. Imaging revealed extensive colocalization between Gars1 mRNA and mitochondria (Fig 3E). To

validate our colocalization approach, we compared the distribution of Gars1 with that of Cox7c mRNA, a transcript with established mitochondrial localization (Cohen et al, 2022). We observed that Gars1 mRNA colocalizes with mitochondria to a comparable extent as Cox7c (Fig S6A).

To determine whether this association is translation-dependent, as reported for other mitochondrial protein mRNAs (Eliyahu et al, 2010; Cohen et al, 2022; Harbauer et al, 2022), we treated cells with cycloheximide or puromycin before fixation. Cycloheximide treatment enhanced colocalization, whereas puromycin significantly reduced it (Fig 3E and F). These results support a translation-dependent recruitment of Gars1 mRNA to

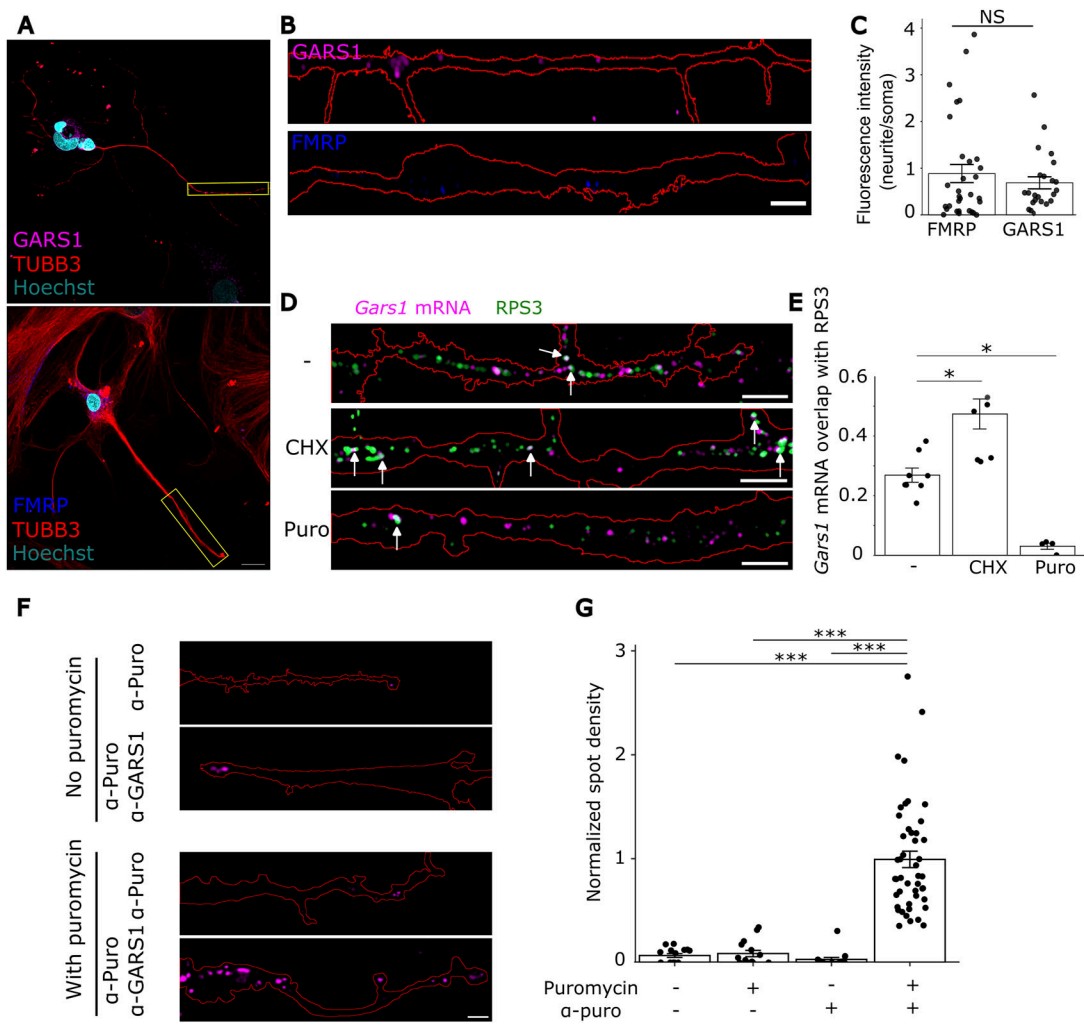

**Figure 2. Gars1 mRNA is locally translated to the GARS1 protein in neurites.**
**(A)** Primary rat cortical neurons (5–7 DIV) were fixed and subjected to immunofluorescence of GARS1 (magenta) and FMRP (blue). Neurites and soma were labeled with α-TUBB3 (red), and nuclei with Hoechst (cyan). Scale bar, 20 μm. **(A, B)** Higher magnification neurite from (A) showing GARS1 and FMRP distribution. Neurite outlines were traced in FIJI. Scale bar, 5 μm. **(C)** Quantification of neurite-to-soma fluorescence intensity. Each dot represents one neurite (n = 23–31 neurites from four biological replicates, ~6 cells per replicate). **(D)** Neurons were treated with cycloheximide (CHX) or puromycin (Puro), or left untreated, followed by smiFISH for Gars1 mRNA (magenta) and RPS3 (green). Neurites were marked with α-TUBB3, and colocalizing puncta (white arrows) were defined as signals with ≥50% overlap of Gars1 and RPS3. Scale bar, 5 μm. **(E)** Quantification of colocalization between Gars1 mRNA and RPS3 clusters. Colocalization was defined as >50% spatial overlap between a Gars1 mRNA and RPS3 (n = 12 cells from four biological replicates, ~3 cells per replicate). Each point represents a cell. **(F)** Cortical neurons were subjected to puromycin proximity ligation assay (Puro-PLA; Tom Dieck et al, 2015), showing representative neurites for each condition. Neurite outlines were drawn from the unthresholded Texas Red channel. Scale bar, 5 μm. **(G)** Normalized neurite-to-soma spot density for each condition. Each point represents a neurite (n = 15–46 neurites per condition from three biological replicates, ~4 cells per replicate). **(C, E, G)** Data information: P-values: *<0.05; ****<0.0001; (NS) > 0.05 by a t test (C); Kruskal–Wallis test followed by pairwise Wilcoxon tests with the Holm–Bonferroni correction (E); Kruskal–Wallis test followed by pairwise Wilcoxon tests with the Bonferroni correction (G). **(C, E, G)** Results are presented as a violin plot (C) with median indicated by a horizontal line; mean ± SEM (E, G).

mitochondria. To further strengthen this possibility, we analyzed the association of Gars1 mRNA with ribosomes through staining of the ribosomal protein RPS3, and mitochondria. We detected Gars1 mRNA puncta that simultaneously overlapped with both ribosomal clusters and mitochondria (Fig 3G). Cycloheximide treatment mildly increased this tripartite association, whereas puromycin treatment significantly reduced it (Fig 3G and H).

Previous work has demonstrated that the coding sequence (CDS) is often sufficient to direct the localization of nuclear-encoded mitochondrial protein mRNAs (Eliyahu et al, 2010;

Williams et al, 2014; Luo et al, 2025). To test whether Gars1 CDS is sufficient for its mitochondrial localization, and to circumvent the limitations of the endogenous probes by explicitly tracking the targeted transcript, we transfected cells with a construct comprising the mitochondrial Gars1 CDS fused to GFP. By detecting the localization of this mRNA using probes against the unique GFP sequence, we could unambiguously identify this specific isoform. We observed extensive colocalization between the GFP mRNA signal (GFP-Cy3) and Mito-BFP (Fig 3I and J), which was not observed with a GFP-only control (Fig S6B). Furthermore, this

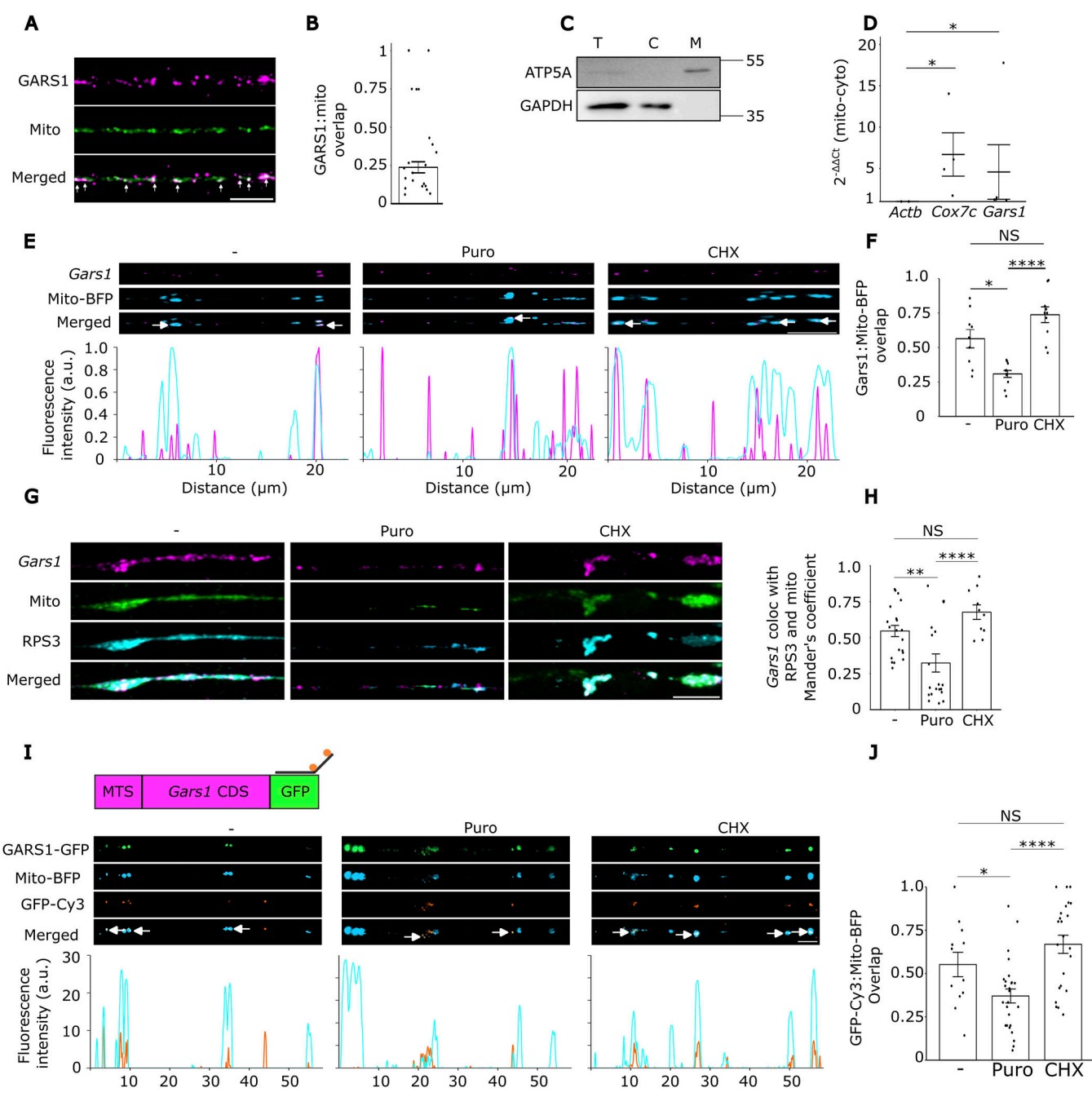

**Figure 3. GARS1 protein and mRNA colocalization with mitochondria.**

N2a cells were seeded and grown to 60–80% confluency, differentiated for 24 h by serum deprivation, and processed as indicated (see the Materials and Methods section). **(A)** Cells were stained with MitoTracker Red (Mito) and immunolabeled for GARS1; white arrows indicate GARS1–mitochondria overlap. Scale bar, 5 $\mu$m. **(A, B)** Quantification of GARS1–mitochondria overlap from panel (A); $n$ = 21 cells from three biological replicates; each dot represents one cell, ~7 cells per replicate. **(C)** Cells were subjected to mitochondrial fractionation followed by protein and RNA extraction from the whole-cell lysate (T), mitochondrial (M), and cytosolic (C) fractions. Protein samples were probed for ATP5A and GAPDH, markers for mitochondria and cytosol, respectively; representative of three biological replicates. **(D)** RT–qPCR analysis of $\beta$-actin (Actb), Cox7c, and Gars1 in mitochondrial versus cytosolic RNA; enrichment was calculated by the $2^{-\Delta\Delta Ct}$ method, where $\Delta Ct = Ct_{Mito} - Ct_{Cyto}$ and $\Delta\Delta Ct$ is normalized to Actb; each dot represents one biological replicate. $n$ = 4–5 biological replicates. **(E)** Cells were transfected with Mito-BFP–expressing plasmid, and subjected to smiFISH for Gars1 mRNA; line scans along neurites illustrate Gars1 mRNA–mitochondria colocalization, with white arrows marking overlap. Scale bar, 5 $\mu$m. **(F)** Quantification of Gars1–Mito-BFP overlap under the indicated conditions; $n$ = 9–11 cells per condition from three biological replicates; each dot represents one cell, ~3 cells per replicate. **(G)** Differentiated N2a cells stained with MitoTracker Red and subjected to smiFISH for Gars1 and IF for RPS3; white arrows indicate Gars1 overlapping with mitochondria and RPS3. Scale bar, 5 $\mu$m. **(E, H)** Mander's coefficients for overlap of Gars1 with mitochondria and RPS3 from panel (E); each dot represents one cell ($n$ = 10–21 cells per condition from three biological replicates, ~3–7 cells per replicate). **(I)** Cells expressing GFP-tagged GARS1 CDS and Mito-BFP were

colocalization was sensitive to puromycin treatment (Fig 3I and J). CHX treatment had a much milder effect, suggesting a saturated ribosomal association under these conditions (Fig 3I and J). Altogether, these data reveal that the Gars1 CDS contains sufficient information to direct translation-dependent mitochondrial association.

### GARS1 protein is in proximity to tRNA$^{Gly}$ in neurites

An abundant localization of GARS1 protein in neurites suggests that it locally binds to and charges its cognate tRNA, tRNA$^{Gly}$. We first tested the presence of tRNAs in neurites, by performing smiFISH for tRNA$^{Gly}$ and tRNA$^{Lys}$. A clear signal was observed for both, whereas no signal was observed in the scrambled control (Fig 4A and B). The signal was present in both soma and neurites, consistent with the necessity for tRNAs for translation in these sites (Fig 4A–C). Furthermore, neurons grown on transwells were fractionated and subjected to RT–qPCR against tRNA$^{Gly}$ and another tRNA (tRNA$^{Ile}$), which substantiated the presence of tRNAs in neurites (Fig 4D).

To test whether GARS1 colocalizes with tRNA$^{Gly}$ in neurites, we performed smiFISH and IF in primary hippocampal neurons and observed an extensive overlap between GARS1 and tRNA$^{Gly}$, and not with the noncognate tRNA$^{Lys}$ (Fig 4E and F). Moreover, the proportion of GARS1 that overlapped with tRNA$^{Gly}$ remained relatively constant across the entire length of a neurite (Fig 4G). We observed similar colocalization patterns in differentiated N2a cells (Fig S7), further strengthening our results. Although the resolution of our microscope cannot definitively resolve direct enzyme–substrate interactions, the specific and persistent spatial proximity between GARS1 and tRNA$^{Gly}$ is consistent with the possibility that tRNA charging occurs locally and facilitates local protein synthesis in neurites.

### Disrupting GARS1 and tRNA$^{Gly}$ proximity impairs neuritic protein synthesis

To assess the functional importance of the GARS1-tRNA$^{Gly}$ interaction in neurites, we aimed to disrupt this association using antisense oligonucleotides (ASOs) and evaluate the impact on local protein synthesis. We designed ASOs that complement the anticodon loop of three tRNA$^{Gly}$ isoacceptors. Introducing these ASOs is expected to sterically hinder GARS1-tRNA$^{Gly}$ interaction (Fig 5A). We conjugated the ASO, and control scrambled sequences to a fluorophore (Cy5) and performed smiFISH. A strong signal in soma and neurites was observed with ASO-Cy5, but not with scramble-Cy5 (Fig S8), indicative of their specificity.

To investigate the importance of the interaction between GARS1 and tRNA$^{Gly}$ in protein synthesis, we transfected the ASOs or the scrambled control into N2a cells, and global protein synthesis

was measured by surface sensing of translation (SUnSET) (Schmidt et al, 2009) followed by Western blot analysis. Although the use of N2a cells is less optimal for aspects of spatial resolution, it allowed efficient transfection of the different ASOs and a sufficient amount of material for Western analysis. While the scrambled control induced a modest reduction in global protein synthesis, presumably because of transfection-related stress, the ASOs caused a more pronounced decrease compared with both the control and untransfected cells (Fig 5B and C). To eliminate the possibility of a reduction in global protein synthesis because of a degradation of tRNA$^{Gly}$, we transfected cells with ASOs or the scrambled control and performed RT–qPCR. We observed no apparent decrease in tRNA levels upon transfection (Fig S9A). We observed no differences in the transfection efficiencies of the ASOs or the scrambled control (Fig S9B), nor did the transfection impair the morphology of the cells (Fig S9C). These results suggest that the ASOs effectively disrupt the interaction between GARS1 and tRNA$^{Gly}$, therefore preventing aminoacylation and hampering global protein synthesis.

We next tested whether the ASOs reduce protein synthesis locally in neurites. To determine the effect of the ASOs on proximity between GARS1 and tRNA$^{Gly}$ in neurites, we transfected N2a cells with Cy5-tagged ASOs or the scrambled control. Then, we performed smiFISH and IF for tRNA$^{Gly}$ and GARS1 protein, respectively (Fig 5D). We observed a dramatic decrease in the amount of tRNA$^{Gly}$ that is in proximity to GARS1 protein in neurites (Fig 5D and E), which suggests that the ASOs effectively disrupt their interaction.

Finally, to determine whether the reduced proximity impairs local protein synthesis, we treated cells with ASOs or scrambled control and labeled nascent protein by puromycin treatment followed by IF with an α-puromycin antibody (Fig 5F and G). The scrambled control led to a slight reduction in protein synthesis in neurites as measured from the puromycin signal (Puro) (Fig 5G and H), presumably because of a general stress induced by the transfection. Nevertheless, the ASOs significantly impaired protein synthesis in neurites (Fig 5H). Altogether, these findings underscore the importance of the proximity between GARS1 protein and tRNA$^{Gly}$ in facilitating local protein synthesis in neurites.

## Discussion

Local protein synthesis in neurites is an intricate mechanism that enables neurons to rapidly regulate their proteome in response to a subset of cues (Cagnetta et al, 2018). Although mRNAs, translation factors, and ribosomes have been identified in neurites, the role of aminoacyl-tRNA synthetases and tRNAs in facilitating local protein synthesis is largely unclear. Here, using smiFISH analysis, we show that Gars1 mRNA is as abundant in the neurites of differentiated neuronal cell lines and primary rat cortical neurons. Moreover, we

---

left untreated or treated with CHX or puromycin and subjected to smiFISH for GFP mRNA; line scans along neurites illustrate colocalization with mitochondria, with white arrows indicating overlap. Scale bar, 5 μm. **(J)** Quantification of GFP mRNA–Mito-BFP overlap under the indicated conditions; $n$ = 10–18 cells per condition from 3 biological replicates, each dot represents one cell, ~3–6 cells per condition per replicate. **(D, F, H, J)** Data information: $P$-values: *<0.05; **<0.01; ****<0.00001; (NS) > 0.05 by the Kruskal–Wallis test followed by Dunn's test with the Bonferroni correction (D); Kruskal–Wallis test followed by Dunn's post hoc test with the Bonferroni correction (F); Kruskal–Wallis test followed by Dunn's post hoc test with the Benjamini–Hochberg correction (H, J). **(D, F, H, J)** Results are presented as the mean ± SEM (D, F, H, J).

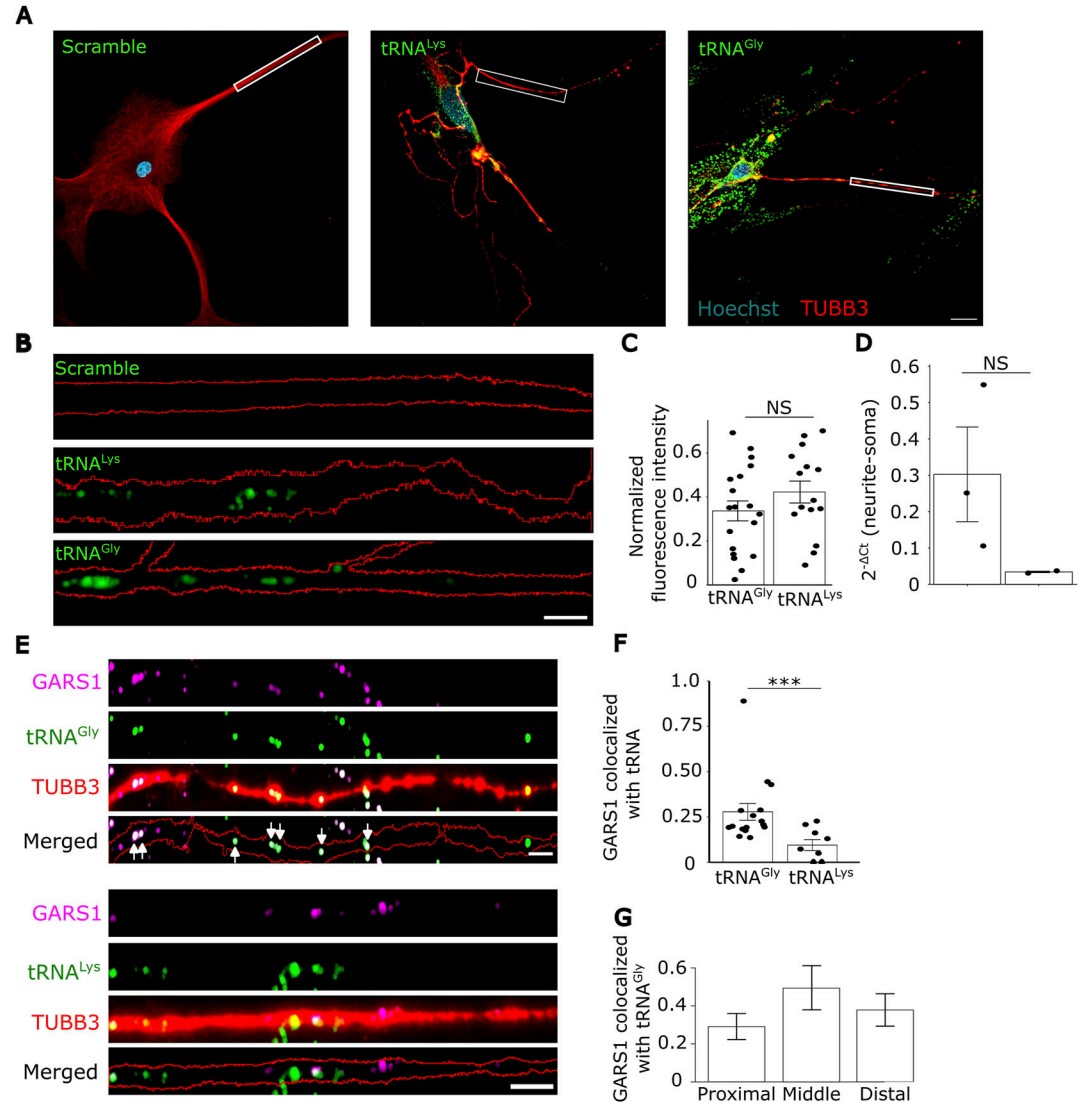

**Figure 4. Colocalization of the GARS1 protein with tRNA.**
**(A)** Primary rat hippocampal neurons were grown for 7–10 d, fixed, and subjected to smiFISH for tRNA[Gly], tRNA[Lys], or scrambled control probes (all tagged with Cy5, shown in green). α-TUBB3 antibody was used to visualize cell bodies and neurites (red), and Hoechst to stain nuclei (cyan). Scale bar, 20 μm. Representative images of tRNA[Gly], tRNA[Lys], or a scrambled control are shown. **(A, B)** Magnification of the neurites indicated in (A) shows the distribution of tRNAs along the neurite. Neurite outline was drawn using FIJI (Schindelin et al, 2012). Scale bar, 5 μm. **(B, C)** Quantification of (B). Mean fluorescence intensity in the soma and neurites was measured and divided by the area of soma and neurite, respectively. Normalized fluorescence intensity was calculated by dividing neurite fluorescence intensity per μm² by soma fluorescence intensity per μm². Each point in the graph represents a neurite (n = 7–11 cells from four biological replicates, ~2–3 cells per replicate). **(D)** CAD cells were seeded at 30% confluency on 1-μm microporous membranes. After 48 h, differentiation was induced by serum starvation (0.8% serum) for 7–14 d. Neurite and cell body fractions were collected and subjected to RNA extraction and RT–qPCR. ΔCt was defined as neurite Ct – soma Ct. n = 2–3 biological replicates. **(E)** Primary rat hippocampal neurons were subjected to smiFISH for tRNA[Gly] and tRNA[Lys] (green), followed by IF with an α-GARS1 antibody (magenta), and an α-TUBB3 antibody to visualize cell body and neurites (red). GARS1 and tRNA puncta that exhibited 50% overlap or higher were considered colocalized. White arrows indicate colocalized tRNA[Gly] and GARS1. Images were acquired by a Zeiss LSM 980 microscope using the Airy Scan 2 SR mode. Scale bar, 5 μm. **(E, F)** Quantification of (E). Each point represents a neurite. n = 3–4 cells per replicate across three biological replicates. **(G)** Distribution of colocalized GARS1-tRNA[Gly] along the neurite. The number of colocalized puncta was quantified in the proximal (first 0–20 μm), middle, and distal (final 20 μm) segments (average neurite length = 110 μm). Each regional count was then expressed as a fraction of the total number of colocalized puncta along the entire neurite. Same samples as in (F). **(C, D, F)** Data information: P-values: ***< 0.001; (NS) > 0.05 by Wilcoxon's test (C, D, F). **(C, D, F, G)** Results are presented as the mean ± SEM (C, D, F, G).

show that Gars1 mRNA is locally translated in neurites. Taken together, these findings suggest that like other locally translated mRNAs that perform a specific function upon their local translation, Gars1 mRNA may be locally translated in neurites to facilitate local protein synthesis. Moreover, like other mRNAs that

are locally regulated upon local cues, localized translation of Gars1 suggests a new, local layer for its expression regulation. The levels of this tRNA-charging enzyme may be coordinated by local needs for translation. The cues and factors that mediate such a local regulation are yet to be uncovered. Moreover, whether

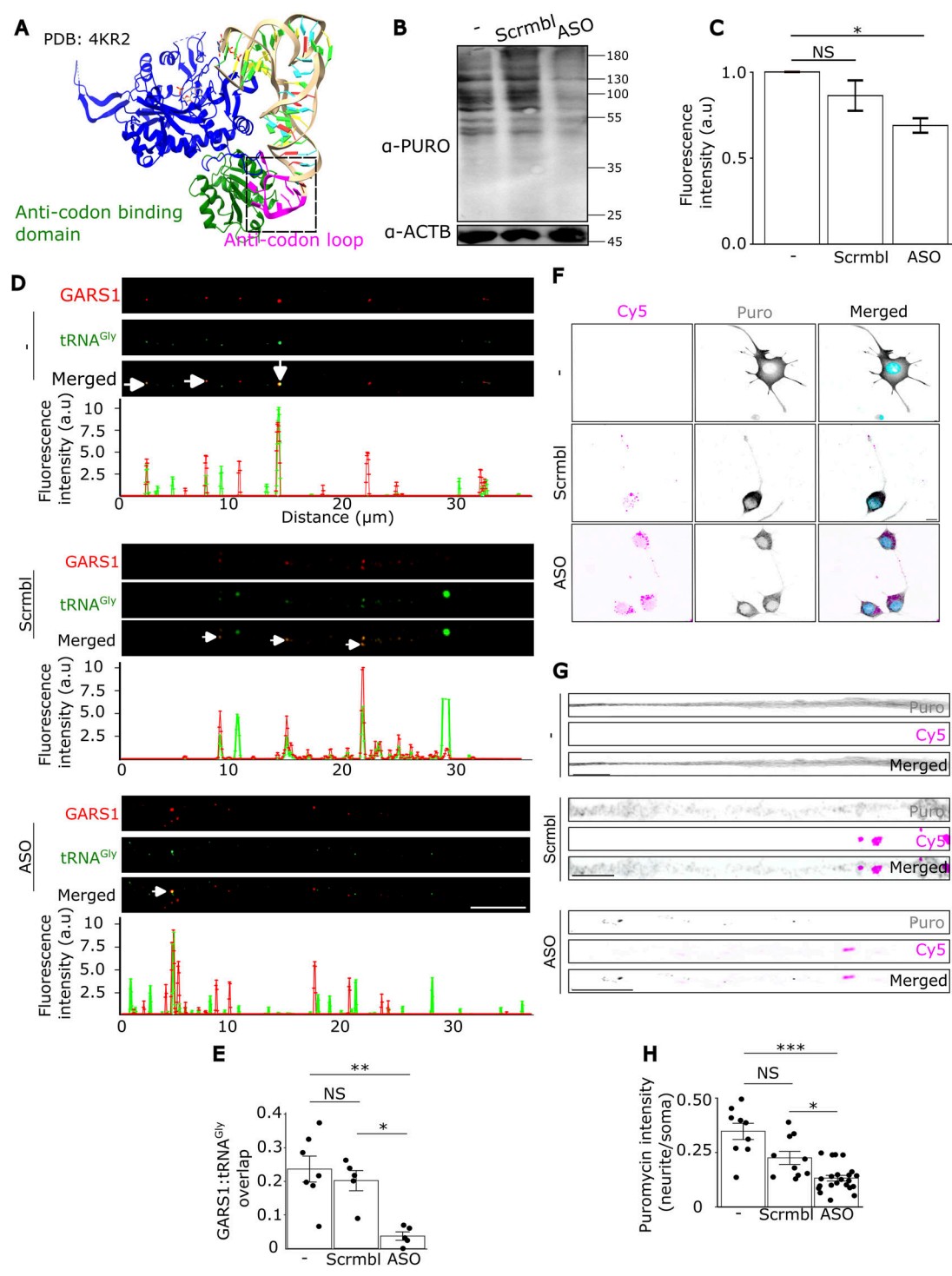

**Figure 5. Disrupting GARS1-tRNA$^{Gly}$ interaction locally affects translation in differentiated N2a cells.**
**(A)** Crystal structure of the human GARS1 protein bound to tRNA$^{Gly}$, visualized on ChimeraX (Cader et al, 2007). PDB: 4KR2. The boxed region (in black) indicates the region the ASOs were designed to hybridize to. **(B)** Cells were subjected to SUnSET (see the Materials and Methods section), followed by lysis and Western blot analysis. A representative image of a single immunoblot is shown using an α-puromycin antibody and an α-ACTB antibody (normalizing protein). **(B, C)** Quantification of puromycin signal (α-Puro) normalized to ACTB (α-ACTB) in (B) from three independent repeats. **(D)** Cells were transfected with ASO, scrambled control, or none, followed by smiFISH for tRNA$^{Gly}$ probes (green), and IF with an α-GARS1 antibody (red). Line scan along the entire neurite indicates the colocalization of tRNA$^{Gly}$ and GARS1 under the different conditions. White arrows indicate colocalized tRNA$^{Gly}$ and GARS1. Scale bar, 5 μm. **(E)** Quantification of tRNA$^{Gly}$ and GARS1 colocalization. Each point represents a cell (n = 3 cells per replicate across three biological replicates). **(F)** Protein synthesis in soma and neurites was measured by IF using an α-puromycin antibody (gray). Transfected cells were identified with a Cy5 fluorophore (magenta) that was appended to the ASO or scrambled control. Nuclei were stained with Hoechst (cyan). Scale bar, 5 μm. **(F, G)** Magnification of the neurites indicated in (F) showing a significant decrease in protein synthesis upon ASO transfection. Scale bar, 5 μm. **(F, G, H)**

Gars1 represents broader phenomena whereby all ARSs are locally translated is yet to be determined. Transcriptomics studies (Minis et al, 2014; Briese et al, 2016; Rotem et al, 2017; Zappulo et al, 2017; Maciel et al, 2018; Middleton et al, 2019), in which other ARSs were also found to be enriched in neurites, suggest that this is the case.

The constant distribution of Gars1 mRNA along neurites suggests that it is localized in distal sites through an active mechanism, which is yet to be revealed. Transport of mRNAs to the distal axons can be accomplished by distinct mechanisms, such as RNA granules attached to kinesin motors (Kanai et al, 2004), or by hitchhiking on organelles, such as lysosomes (Liao et al, 2019), endosomes (Cioni et al, 2019; Corradi et al, 2020), or mitochondria (Cohen et al, 2022; Harbauer et al, 2022). Our observation that Gars1 mRNA is associated with mitochondria (Fig 3E) supports the latter possibility. Nevertheless, Gars1 mRNA was among many mRNAs found associated with P180 receptor, which is a component of the axonal ER tubules (Koppers et al, 2024). This may suggest an additional mode of transport, via axonal ER tubules.

Elucidating the interaction between aminoacyl-tRNA synthetases and their cognate tRNAs is important for understanding how mutations in these domains lead to genetic and neurological diseases (Antonellis & Green, 2008) and whether defects in local protein synthesis lead to neurological disorders (Lin et al, 2021). Specifically, mutations in the catalytic and anticodon binding domains of the GARS1 protein have been shown to cause Charcot–Marie–Tooth type 2D disease and other peripheral neuropathies (Antonellis et al, 2003). These mutations expose protein domains that induce aberrant protein interactions, suggesting a gain-of-function mechanism (He et al, 2015; Mo et al, 2018). Yet, mutations in GARS1 protein's catalytic domains were also shown to impair global protein synthesis (Niehues et al, 2015). This impairment may have been due to impaired GARS1-tRNA$^{Gly}$ interactions, because the overexpression of tRNA$^{Gly}$ restored a normal phenotype in CMT2D mouse models (Zuko et al, 2021). Here, by analyzing changes in proximity between GARS1 and its cognate tRNA we examine the outcomes of local interaction between them in facilitating local protein synthesis. It remains to be investigated whether GARS1 mutations that appeared to affect global protein synthesis also impair local protein synthesis. Investigating this will provide important hints regarding the molecular mechanisms underlying the peripheral phenotypes induced by these mutations. Moreover, GARS1 mutations that generate aberrant protein interactions are associated with defects in retrograde transport in axons (Sleigh et al, 2023); it is yet to be determined whether impaired localized translation underlies this transport problem.

The data presented here support a model in which Gars1 mRNA is colocalized with mitochondria in neurites and is locally translated near the vicinity of mitochondria, which might facilitate the import of the GARS1 protein into mitochondria. Nuclear-encoded mitochondrial protein mRNAs were previously shown to associate with mitochondria in a translation-dependent manner (Eliyahu et al, 2010; Williams et al, 2014; Fazal et al, 2019), and the current results extend this concept by positioning Gars1 mRNA as a dual-function transcript that supports both mitochondrial and cytosolic translation demands in neurites. It was previously shown that the localization of the GARS1 protein is coordinated through two transcripts, one which contains an upstream ORF (uORF) that leads to the translation of an MTS-devoid protein (i.e., cytosolic protein) and the other that encodes an MTS-containing transcript which translates the mitochondrial variant of GARS1 (Alexandrova et al, 2015). It is therefore likely that the mitochondria-localized Gars1 mRNA encodes the mitochondrial enzyme and translates the MTS. Consistent with that, introducing a plasmid expressing GARS1 with an MTS (Fig 3I) showed mRNA localization to mitochondria. Nevertheless, we cannot exclude the possibility that the other transcript is also mitochondria-associated and translated the cytosolic variant on the outer mitochondrial membrane. This will substantiate the idea that the mitochondria serve as a hub for translation, not necessarily of mitochondrial proteins.

In summary, this study shows that the essential components of the translational machinery, namely, Gars1 mRNA and protein, are spatially organized with mitochondria to support local protein synthesis. Gars1 mRNA associates with mitochondria in a translation-dependent manner, and is locally translated. The resulting GARS1 protein binds and charges tRNA$^{Gly}$ in neurites, and disrupting this interaction impairs both global and local protein syntheses.

# Materials and Methods

### Rat primary neuronal culture

All animal experiments were performed in accordance with relevant guidelines and regulations of the Technion Animal Ethics Committee (Permit #IL-039-03-25). Primary cortical and hippocampal neuronal cultures were prepared according to Berlin and Isacoff (2018) with minor alterations. For each experiment, 2 postnatal 1- to 4-d (P1–P4) pups were ordered and euthanized by decapitation under approved protocols upon arrival. Brains were rapidly removed into a petri dish containing prewarmed (37°C) dissection/dissociation (D/D) medium (HBSS (Ca$^{2+}$ and Mg$^{2+}$ free), 10 mM Hepes, 20 mM glucose, pH 7.4). Under a dissecting microscope, meninges were peeled away, hippocampi and cortices were isolated intact, and each tissue was transferred into a 15-ml conical tube containing prewarmed D/D medium. Tissues were dissociated by 15-min incubation in 0.25% trypsin at 37°C, then rinsed six times in D/D medium to remove trypsin and residual meninges. After the final rinse, D/D medium was removed, and tissues were

---

Quantification of puromycin fluorescence intensity in neurites (normalized to soma) in (F, G). Each dot represents a single cell (*n* = 10–26 cells, ~3–8 cells per biological replicate). All experiments in this figure were performed in *n* = 3 biological replicates. **(C, E, H)** Data information: *P*-values: *<0.05; **<0.01; ***< 0.0001; (NS) > 0.05 by a priori *t* test (C); Kruskal–Wallis test followed by Dunn's test with the Holm correction (E); Kruskal–Wallis test followed by Dunn's test with the Bonferroni correction (H). **(C, E, H)** Results are presented as the mean ± SEM (C, E, H).

resuspended in 1 ml neuronal growth medium (NGM: 25 ml fetal bovine serum, 10 ml NeuroCult SM1 Neuronal Supplement [Cat. No. 05711; STEMCELL Technologies], 2 mM L-glutamine, 20 mM d-glucose, 50 $\mu$l serum extender [Cat. No. 355006; BD Biosciences] in MEM, final volume 500 ml), triturated six times through a fire-polished glass pipette, and transferred to a cell strainer. This step was repeated three times, to obtain a single-cell suspension.

Neurons were plated onto poly-D-lysine–coated glass coverslips in 24-well plates at 100,000 cells/well and maintained at 37°C in a humidified 5% $CO_2$ incubator. 5 d after plating, half of the medium was replaced with NGM supplemented with cytosine arabinoside (AraC, final concentration 4 $\mu$M) to suppress glial proliferation, and then, medium (NGM supplemented with 4 $\mu$M AraC) was changed every 2–3 d. Neurons were used for experiments between 7 and 14 d in vitro.

### Cell culture

Neuro2a (N2a) cells were grown in DMEM, supplemented with 10% FBS, 1% L-glutamine, and 0.1% penicillin–streptomycin. The culture environment was maintained at a constant temperature of 37°C with a 5% $CO_2$ atmosphere. The cells were passaged every 3–4 d at 30% confluency by trypsinization. Serum deprivation was performed to induce differentiation by incubating cells with differentiation media (1% L-glutamine and 0.1% penicillin–streptomycin in DMEM) for 24 h. Cath.a-differentiated (CAD) cells were grown in DMEM:F12, supplemented with 8% FBS, 1% L-glutamine, and 0.1% penicillin–streptomycin, on poly-D-lysine–coated plates. CAD cells were grown and passaged under the same conditions as N2a cells. Serum deprivation was performed to induce differentiation by incubating cells with differentiation media (0.8% FBS, 1% L-glutamine, and 0.1% penicillin–streptomycin in DMEM) for 7 d. For local translation analysis, puromycin (200 $\mu$g/ml) or cycloheximide (250 $\mu$g/ml) was added to the fully supplemented medium for 45 or 30 min, respectively.

Transient transfection was performed using jetPRIME transfection reagent (Polyplus-transfection) according to the manufacturer's instructions.

Neurite fractionation was performed according to Arora et al (2021) and Cohen et al (2022). Briefly, cells were seeded on poly-D-lysine–coated porous membranes with 1-$\mu$m pores and grown for 7–14 d. Neurons were fractionated by gently scraping the axonal fraction from the bottom of the membrane and the cell bodies from the top of the membrane. The two fractions were collected into separate tubes and used for RNA extraction and RT–qPCR analysis.

### Plasmid construction

The coding sequence (CDS) of the human GARS1 gene (NM_002047), including its MTS, was cloned into a pEGFP-N3 mammalian expression vector, generating a C-terminally tagged GARS1 expressed under CMV promoter.

The RPL10a-GFP expression plasmid was kindly provided by the laboratory of Prof. Eran Perlson (Tel Aviv University, Israel).

### Reverse transcription followed by quantitative PCR (RT–qPCR)

Reverse transcription of Gars1 and Actb mRNA was performed using PrimeScript RT Reagent Kit with gDNA eraser (RR047Q; Takara Bio) according to the manufacturer's instructions. Reverse transcription of tRNA samples was performed according to Levi et al (2023) using RevertAid First Strand cDNA Synthesis Kit (Thermo Fisher Scientific).

qPCR was done in a 20 $\mu$l reaction volume in triplicates using Fast SYBR Green PCR Master Mix (Cat. No. 4367659; Applied Biosystems) following the manufacturer's instructions using primers for the indicated genes (Table 1). All qPCR parameters were used in accordance with Levi and Arava (2021). Results were analyzed using Microsoft Excel and R software. $\Delta C_t$ values were calculated by taking the difference of the mean $C_t$ for the soma and neurite samples (Neurite $C_t$-Soma $C_t$).

### smiFISH

Gars1 mRNA probe sets (24 probes, Table 2) were designed using Stellaris Probe Designer (masking level: 5, oligo length: 22, minimum spacing; LGC Biosearch Technologies). tRNA probes were manually designed to interact with accessible sites of the mature tRNA (tRNA$^{Gly}$-GCC, tRNA$^{Gly}$-UCC, and tRNA$^{Gly}$-CCC, tRNA$^{Lys}$-CUU, and tRNA$^{Lys}$-UUU). All probes were aligned to the mouse and rat genomes at UCSC Genome Browser (Kent et al, 2002) to ensure no off-target binding. Then, GtRNAdb (Chan & Lowe, 2016) was used to validate that the probe sequences that correspond to their genomic sequence correspond to those of mature tRNAs. Probes were annealed to either a FLAP-X or a FLAP-Y tagged with a Cy3 or a Cy5 fluorophore, respectively, according to Tsanov et al (2016).

smiFISH was performed on primary cells grown as described above (Rat primary neuronal culture section), or cell lines grown to 60% confluency on coverslips in a 24-well plate, then differentiated for 24 h. Cells were fixed with 4% PFA in PBS for 10 min at room temperature (RT), then permeabilized with 0.1% Triton X-100 in 70% ethanol overnight at 4°C. After overnight incubation, cells were incubated with a fresh formamide buffer (10% formamide, 2× SSC in RNase-free water) for 15 min at RT. Each coverslip was placed face down on a 50 $\mu$l drop of hybridization buffer (containing the annealed probes) and incubated overnight at 37°C in a humidified chamber. After hybridization, cells were washed twice with a fresh and prewarmed (37°C) formamide buffer for 30 min at 37°C. Cells were washed twice with PBS, 5 min per wash. Hoechst staining was conducted by placing the coverslip face down on a 50 $\mu$l drop of 2 $\mu$g/ml Hoechst in PBS for 5 min at RT. Then, samples were washed once with PBS for 5 min at RT, mounted with Fluoromount-G, and allowed to dry for at least 1 h. Finally, coverslips were sealed by a nail polish and kept at 4°C.

### Immunofluorescence

Immunofluorescence was performed on primary cells grown as described above (Rat primary neuronal culture section), or cell lines grown to 60% confluency on coverslips in a 24-well plate, then differentiated for 24 h. Cells were rinsed twice in PBS, then fixed with 4% PFA in PBS for 10 min at RT. Cells were permeabilized by

**Table 1. Primers.**

| Primer name | Laboratory number | Sequence (5'-3') |
| --- | --- | --- |
| Gars1_qPCR_F | 380 | AGACCATCCCAAGTTCCAAAG |
| Gars1_qPCR_R | 381 | TTCAACAGCATCTCCCAGAC |
| ActB_qPCR_F (from Cohen et al [2022]) | 238 | ACCTTCTACAATGAGCTGCG |
| ActB_qPCR_R (from Cohen et al [2022]) | 239 | CTGGATGGCTACGTACATGG |
| tRNA-Gly-GCC-1-1_cDNA_forward | 382 | GCATGGGTGGTTCAGTGG |
| tRNA-Gly-GCC-1-1_cDNA_reverse | 383 | TGCATGGGCCGGGAATCGAA |
| tRNA-Ile-AAU-1-1_forward | 384 | GGCCGGTTAGCTCAGTTG |
| tRNA-Ile-AAU-1-reverse | 385 | CCGTACGGGGATCGAA |
| Cox7c_mouse_F | 360 | CATGTTGGGCCAGAGTATC |
| Cox7c_mouse_R | 361 | AACCCAGATCCAAAGTACACG |

0.1% Triton X-100 in PBS for 10 min at RT and blocked by blocking solution (1% BSA, 0.1% Triton X-100 in PBS) for 1 h at RT. Each coverslip was then placed face down on on a 30 µl drop of the diluted primary antibody (Table 3) solution in a blocking solution and incubated for 1 h at RT. Coverslips were transferred to a 24-well plate and were washed three times with PBS, 5 min per wash, with rocking. Each coverslip was placed face down on  on a 30 µl drop of secondary antibody (diluted 1:500 in a blocking solution) and incubated for 1 h at RT in the dark. Cells were washed twice with PBS, 5 min per wash. Hoechst staining was conducted by placing the coverslip face down on a 50 µl drop of 2 µg/ml Hoechst in PBS for 5 min at RT. Then, samples were washed once with PBS for 5 min at RT, mounted with Fluoromount-G, and allowed to dry for at least 1 h. Finally, coverslips were sealed with a nail polish and kept at 4°C.

In cases where IF was performed after smiFISH, cells were repermeabilized with 0.1% Triton X-100 for 10 min at RT followed by 30-min incubation in blocking solution at 37°C. The rest of the IF protocol was carried out as described above, except that primary and secondary antibody incubations were performed at 37°C.

## Antisense oligo (ASO) treatment

Single-stranded DNA oligos (22 nts) complementary to the anti-codon loop of mature tRNA$^{Gly}$-GCC, tRNA$^{Gly}$-UCC, and tRNA$^{Gly}$-CCC were manually designed. As a control, these sequences were scrambled (Table 4). Each probe was aligned to the rat and mouse genomes at UCSC Genome Browser (Kent et al, 2002) to exclude off-target binding, and binding to the mature tRNA was validated at GtRNAdb (Chan & Lowe, 2016). The FLAP-Y sequence (Tsanov et al, 2016) was added to the 5' end of each probe. The individual probes were then annealed to a complementary FLAP-Y sequence tagged with a Cy5 fluorophore on the 5' and 3' termini according to Tsanov et al (2016), resulting in ASO-Cy5 (ASO) or scramble-Cy5 (Scrmbl). These oligos were then transfected with jetPRIME at a final concentration of 1 µM for 24 h to N2a cells at 60–80% confluency, followed by differentiation for 24 h. smiFISH and IF were then performed to assess the colocalization of the GARS1 protein and tRNA$^{Gly}$.

## SUnSET

N2a cells at 60% confluency were transfected with the ASO or scrambled oligos at a final concentration of 1 µM for 24 h using jetPRIME transfection reagent. Cells were then differentiated for 24 h, and SUnSET was performed to determine the effect of ASOs on translation (Schmidt et al, 2009). Briefly, samples were washed twice with warm PBS, incubated with puromycin-containing media (5 µg/ml puromycin) for 15 min, then washed again twice with warm PBS. Cells were scraped and lysed in 1 ml lysis buffer (20 mM Tris–HCl, 150 mM NaCl, 5 mM MgCl$_2$, 2% Triton X-100, 0.1 mM DTT, 1 mM PMSF, 2 µg/ml aprotinin, 10 µg/ml leupeptin, 1 µg/ml pepstatin A), then placed on ice for 20 min, and centrifuged at 14,000$g$ for 15 min at 4°C. The supernatant was mixed with SDS loading buffer and subjected to Western blot analysis (Eliyahu et al, 2010) to measure protein synthesis.

## Mitochondrial fractionation

Mitochondrial fractionation was performed as described in Cohen et al (2022). All steps were conducted on ice or at 4°C using prechilled centrifuges and solutions. Cells (usually $10^7$) were washed twice with ice-cold PBS, detached from the plate using a cell scraper, and collected into a 15-ml conical tube, followed by centrifugation at 600$g$ for 5 min at 4°C to pellet intact cells.

The cell pellet was resuspended in 1 ml of homogenization buffer (0.6 M mannitol, 50 mM Tris–HCl, pH 7.4, 5 mM MgAc, 100 mM KCl, 1 mM DTT, 1 g/liter BSA, 200 µg/ml CHX, 1 mM PMSF, 2 µg/ml aprotinin, 10 µg/ml leupeptin, 1 µg/ml pepstatin A) and chilled on ice for 5 min before transferring to a glass Dounce homogenizer (tight-fitting "B" pestle for low volumes). Homogenization was achieved by 17–20 strokes, followed by a 15-min incubation on ice to ensure complete lysis. The homogenate was centrifuged at 1,000$g$ for 5 min at 4°C, and the supernatant (containing the organelles) was transferred to a new tube.

One-third (300 µl) of the postnuclear supernatant was reserved as the total (T) sample. From this, 10% was aliquoted for protein analysis and the remainder was subjected to RNA extraction. The remaining lysate (~600 µl) was centrifuged at 15,000$g$ for 15 min at 4°C to separate the cytosolic supernatant (C fraction, including

**Table 2. smiFISH probe sequences.**

| Probe name | Sequence (*FLAP-Y is appended at the 5' end*) |
|---|---|
| Gars1_1 | GTAGAAAAACCTCCTCTTCAAC |
| Gars1_2 | CTCCATAAATAGCAAAAGCCTG |
| Gars1_3 | CGAAGTCATACAATCCACTGAC |
| Gars1_4 | GATCTGCTCCTCTTGGATAAAG |
| Gars1_5 | AGAGGTCTTTAAAACTGGCTCA |
| Gars1_6 | ATAGTTATCAAGCTGGGCCAAG |
| Gars1_7 | TAAAAGGTACCGGAGGGGACAG |
| Gars1_8 | CCAATGAAGGTCTGGAACATTA |
| Gars1_9 | CTCAGATATCCAGGCATATTTC |
| Gars1_10 | TTTCTCAGTGGGATCTACAAAG |
| Gars1_11 | CACACTTTGGAACTTGGGATGG |
| Gars1_12 | TTAATCACACCCTGTTCAACAG |
| Gars1_13 | CTTCGTGAGGTAGAGGTAGATG |
| Gars1_14 | CGGAGTTTATCAGGAGATATTC |
| Gars1_15 | CTCAGCTACTAGTGGAACTTTG |
| Gars1_16 | TTGTTGGGCTCAAACTGTACAA |
| Gars1_17 | AAATGTAGCACTCATCACAGGC |
| Gars1_18 | TCTCGGACATGGAATGTATGTT |
| Gars1_19 | AACTGAAGAACGTTCTCTGTTC |
| Gars1_20 | TCAGTGGAAGGACAGAACATTT |
| Gars1_21 | AATAGTGATGCCGAAAGCCACG |
| Gars1_22 | CGTCTTGTTCACTGTATCAAAG |
| Gars1_23 | TATTAGCACGATGGTCATAAGC |
| Gars1_24 | GTGTAGTGGACATGATGAGTTA |
| tRNA-Gly-CCC-2-1_probe 1 | TCTACCATTGAACTACCAATGC |
| tRNA-Gly-CCC-2-1_probe 2 | TCACCCGCGTGGGAGGCGAGAA |
| tRNA-Gly-CCC-2-1_probe 3 | CATTGGCCGGGAATCGAACCCG |
| tRNA-Gly-GCC-1-1_probe 1 | TCTACCACTGAACCACCCATGC |
| tRNA-Gly-GCC-1-1_probe 2 | CCTCCCGCGTGGCAGGCGAGAA |
| tRNA-Gly-GCC-1-1_probe 3 | CATGGGCCGGGAATCGAACCCG |
| tRNA-Gly-UCC-1-1_probe 1 | CTCACCACTATACCACCAACGC |
| tRNA-Gly-UCC-1-1_probe 2 | CAACTGCTTGGAAGGCAGCTAT |
| tRNA-Gly-UCC-1-1_probe 3 | CGTTGGCCGGGAATCGAACCCG |
| Scrambled_1 | CCGCACATGACTCTTAAATTAC |
| Scrambled_2 | ACGGTCGCCCAGCTGGAAAGGG |
| Scrambled_3 | GCTTAGCTCCCGGGCAAGGAAC |
| tRNA_Lys_CUU_12_1_1 | TCTACAGACTGAGCTAGCTGGG |
| tRNA_Lys_CUU_12_1_2 | ACCCTGAGACTAAGAGTCTCAT |
| tRNA_Lys_CUU_12_1_3 | CACAACATGGGGCTCCAACCCA |
| tRNA-Lys-UUU-1-1_1 | CTACCGACTGAGCTATCCGGGC |
| tRNA-Lys-UUU-1-1_2 | CCCTCAGATTAAAAGTCTGATG |

**Table 2. Continued**

| Probe name | Sequence (*FLAP-Y is appended at the 5' end*) |
|---|---|
| tRNA-Lys-UUU-1-1_3 | CCGAACAGGGACTTGAACCCTG |
| Cox7c-1 | ATCATAGCCAGCAACCGCCACTTGTT |
| Cox7c-2 | TCATAGTGGCTGCGACGGACCACGGA |
| Cox7c-3 | CTTGCTCGGCAGAGCGCGAAGACCGA |
| Cox7c-4 | TCAGATCATCTCTTAAACTTTCTTCATTCTGT |
| Cox7c-5 | CGGACCACGGAGGTCGTGAACCTCCG |
| Cox7c-6 | CTGGAAGTTCTGCGAGGGCCGCAGAC |
| Cox7c-7 | CCTTTCTACACGACCTTGCTCGGCAG |
| Cox7c-8 | TAGCTGGTGTCTTACTATAAAGAAAGGTGCGG |
| Cox7c-9 | AGAAAGGTGCGGCAAACCCAGATCCAAAG |
| Cox7c-10 | CCAGATCCAAAGTACACGGTCATCATAGC |
| Cox7c-11 | GGCCCAACATGTCGCTGCTGGAAGTT |
| Cox7c-12 | AGGGCCGCAGACACGGAAGGCGGAAG |
| Cox7c-13 | GGAAGGCGGAAGAAATGGCCGTACCACC |
| Cox7c-14 | CGTACCACCTAACTCCCCTTTCTACA |
| Cox7c-15 | CCGCCACTTGTTTTCCACTGAAAATGGC |
| Cox7c-16 | CTGAAAATGGCAAATTCTTCCCCGGACC |

| Probe name | Sequence (FLAP-X is appended at the 5' end) |
|---|---|
| tRNA-Gly-CCC-2-1_probe 1 | TCTACCATTGAACTACCAATGC |
| tRNA-Gly-CCC-2-1_probe 2 | TCACCCGCGTGGGAGGCGAGAA |
| tRNA-Gly-CCC-2-1_probe 3 | CATTGGCCGGGAATCGAACCCG |
| tRNA-Gly-GCC-1-1_probe 1 | TCTACCACTGAACCACCCATGC |
| tRNA-Gly-GCC-1-1_probe 2 | CCTCCCGCGTGGCAGGCGAGAA |
| tRNA-Gly-GCC-1-1_probe 3 | CATGGGCCGGGAATCGAACCCG |
| tRNA-Gly-UCC-1-1_probe 1 | CTCACCACTATACCACCAACGC |
| tRNA-Gly-UCC-1-1_probe 2 | CAACTGCTTGGAAGGCAGCTAT |
| tRNA-Gly-UCC-1-1_probe 3 | CGTTGGCCGGGAATCGAACCCG |
| mActinbeta_1 | GGAGGGTGAGGGACTTCCTGTAACCACTTATCCT |
| mActinbeta_2 | AAGTCCTCAGCCACATTTGTAGAACTTTGGGGCCT |
| mActinbeta_3 | ACATCTGCTGGAAGGTGGACAGTGAGGCCCCT |
| mActinbeta_4 | GCGCTCAGGAGGAGCAATGATCTTGATCTTCCT |
| mActinbeta_5 | TGGTGCTAGGAGCCAGAGCAGTAATCTCCCCT |
| mActinbeta_6 | ATGGTGGTACCACCAGACAGCACTGTGTTCCT |

| Probe name | Sequence (FLAP-X is appended at the 5′ end) |
|---|---|
| mActinbeta_7 | AGAGGTCTTTACGGATGTCAACGTCACACTTCCCT |
| mActinbeta_8 | TGAATGTAGTTTCATGGATGCCACAGGATTCCCCT |
| mActinbeta_9 | GCATCGGAACCGCTCGTTGCCAATAGTGACCT |
| mActinbeta_10 | TCTCCTGCTCGAAGTCTAGAGCAACATAGCCCT |
| mActinbeta_11 | CGTGAGGGAGAGCATAGCCCTCGTAGATGGCCT |
| mActinbeta_12 | CCATCACAATGCCTGTGGTACGACCAGAGCCT |
| mActinbeta_13 | CCCAGTTGGTAACAATGCCATGTTCAATGGGGCCT |
| mActinbeta_14 | CAGCGATATCGTCATCCATGGCGAACTGGCCT |
| mActinbeta_15 | CTGCATCCTGTCAGCAATGCCTGGGTCCT |
| mActinbeta_16 | CTGGCCGTCAGGCAGCTCATAGCTCTCCT |
| mActinbeta_17 | AGCTTCTCTTTGATGTCACGCACGATTTCCCCT |
| mActinbeta_18 | CTCAGCTGTGGTGGTGAAGCTGTAGCCACCCT |
| mActinbeta_19 | TCGGTCAGGATCTTCATGAGGTAGTCTGTCAGCCT |
| mActinbeta_20 | CGGCCAGCCAGGTCCAGACGCAGGATCCT |
| mActinbeta_21 | CGTACATGGCTGGGGTGTTGAAGGTCTCAAACCCT |
| mActinbeta_22 | GATCTGGGTCATCTTTTCACGGTTGGCCTTCCT |
| mActinbeta_23 | GGTTCAGGGGGGCCTCGGTGAGCAGCCCT |
| Tubb3_1 | TCACTCGGCTAGAGACGAAGAG |
| Tubb3_2 | CTCATTGTAGTAGACACTGATG |
| Tubb3_3 | GACCGAAGATAAAGTTGTCAGG |
| Tubb3_4 | AGTCACAATTCTCACATTCTTT |
| Tubb3_5 | CAGTGAGTGAGTGAGCTGGAAG |
| Tubb3_6 | CACGCTGAAGGTGTTCATGATG |
| Tubb3_7 | TAGCAGACACAAGGTGGTTGAG |
| Tubb3_8 | TGAGCTGACCAGGGAATCGAAG |
| Tubb3_9 | CAAAGCCGGGCATGAAGAAATG |
| Tubb3_10 | AGTAACTACTGTTCTTACTCTG |
| Tubb3_11 | ATGAAGGTGGACGACATCTTGA |
| Tubb3_12 | GAAGAGCTGGTAGGAAGGTAAG |
| Tubb3_13 | AATACTGAAGGACTTTAACCCG |
| Tubb3_14 | AAAACGGGGAGGACAGAGCCAA |
| Tubb3_15 | GCAACATAAATACAAAGGTGGC |
| Tubb3_16 | CCTGGAGCCATAATAAAACAGA |
| Tubb3_17 | CTCCAAATATAAACACAACCCA |
| Tubb3_18 | AGTGACTTTATTAAGTAGACCC |
| Tubb3_19 | TTTTTTTTTTATGTGACAGACA |
| GFP_probe_1 | CCTCCTAAGTTTCGAGCTGGACTCAGTGTTACTTGTACAGCTCGTCCA |
| GFP_probe_2 | CCTCCTAAGTTTCGAGCTGGACTCAGTGAGAGTGATCCCGGCGGCGGT |
| GFP_probe_3 | CCTCCTAAGTTTCGAGCTGGACTCAGTGCTTCTCGTTGGGGTCTTTGC |

| Probe name | Sequence (FLAP-X is appended at the 5′ end) |
|---|---|
| GFP_probe_4 | CCTCCTAAGTTTCGAGCTGGACTCAGTGGCGGACTGGGTGCTCAGGTA |
| GFP_probe_5 | CCTCCTAAGTTTCGAGCTGGACTCAGTGTGTCGGGCAGCAGCACGGGG |
| GFP_probe_6 | CCTCCTAAGTTTCGAGCTGGACTCAGTGGCCGATGGGGGTGTTCTGCT |
| GFP_probe_7 | CCTCCTAAGTTTCGAGCTGGACTCAGTGTGGTCGGCGAGCTGCACGCT |
| GFP_probe_8 | CCTCCTAAGTTTCGAGCTGGACTCAGTGCCTCGATGTTGTGGCGGATC |
| GFP_probe_9 | CCTCCTAAGTTTCGAGCTGGACTCAGTGGTTCACCTTGATGCCGTTCT |
| GFP_probe_10 | CCTCCTAAGTTTCGAGCTGGACTCAGTGTCTGCTTGTCGGCCATGATA |
| GFP_probe_11 | CCTCCTAAGTTTCGAGCTGGACTCAGTGTAGACGTTGTGGCTGTTGTA |
| GFP_probe_12 | CCTCCTAAGTTTCGAGCTGGACTCAGTGGTTGTACTCCAGCTTGTGCC |
| GFP_probe_13 | CCTCCTAAGTTTCGAGCTGGACTCAGTGCCAGGATGTTGCCGTCCTCC |
| GFP_probe_14 | CCTCCTAAGTTTCGAGCTGGACTCAGTGTTGAAGTCGATGCCCTTCAG |
| GFP_probe_15 | CCTCCTAAGTTTCGAGCTGGACTCAGTGCTCGATGCGGTTCACCAGGG |
| GFP_probe_16 | CCTCCTAAGTTTCGAGCTGGACTCAGTGTGTCGCCCTCGAACTTCACC |
| GFP_probe_17 | CCTCCTAAGTTTCGAGCTGGACTCAGTGTCGGCGCGGGTCTTGTAGTT |
| GFP_probe_18 | CCTCCTAAGTTTCGAGCTGGACTCAGTGGCCGTCGTCCTTGAAGAAGA |
| GFP_probe_19 | CCTCCTAAGTTTCGAGCTGGACTCAGTGTGGTGCTCTTCATCTTGTTG |
| GFP_probe_20 | CCTCCTAAGTTTCGAGCTGGACTCAGTGGGGTAGCGGCTGAAGCACTG |
| GFP_probe_21 | CCTCCTAAGTTTCGAGCTGGACTCAGTGCACGCCGTAGGTCAGGGTGG |
| GFP_probe_22 | CCTCCTAAGTTTCGAGCTGGACTCAGTGCGGCGGTCACGAACTCCAGC |
| GFP_probe_23 | CCTCCTAAGTTTCGAGCTGGACTCAGTGGTCGCCGTCCAGCTCGACCA |
| GFP_probe_24 | CCTCCTAAGTTTCGAGCTGGACTCAGTGAACAGCTCCTCGCCCTTGCT |
| FLAP-X (Tsanov et al, 2016) | /Cy3/CACTGAG TCCAGCTCGAAACTTAGGAGG/Cy3/ |
| FLAP-Y (Tsanov et al, 2016) | /Cy5/AATGCATGTCGACGAGGTCCGAGTGTAA/Cy5/ |

soluble components and light organelles) from the crude mitochondrial pellet. The pellet was resuspended in homogenization buffer and centrifuged again at 15,000*g* for 15 min at 4°C to remove residual cytosol, and the final enriched mitochondrial pellet

**Table 3.   Antibodies.**

| Antibody | Source | Identifier | Dilution |
|---|---|---|---|
| ATP5A | Abcam | ab14748 | 1:1,000 |
| βIII-tubulin | Abcam | ab52623 | 1:250 |
| βIII-tubulin | Developmental Studies Hybridoma Bank (DSHB) | 6G7 | 1:200 |
| FMRP | Abcam | ab17722 | 1:100 |
| GAPDH | Abcam | ab181602 | 1:2,000 |
| GARS1 | Santa Cruz Biotechnology | sc-365311 | 1:100 |
| MAP2 | Cell Signaling | #4542 | 1:250 |
| NEFM | Abcam | ab7794 | 1:250 |
| Puromycin | Abcam | ab315887 | 1:100 |
| Puromycin | Developmental Studies Hybridoma Bank (DSHB) | PMY-2A4 | IF 1:10; Western 1:100 |
| RPS3 | Cell Signaling Technology | #9538 | 1:100 |

**Table 4.   ASO sequences.**

| ASO | Sequence |
|---|---|
| tRNA-Gly-CCC-2-1_ASO | TCACCCGCGTGGGAGGCGAGAA |
| tRNA-Gly-GCC-1-1_ASO | CCTCCCGCGTGGCAGGCGAGAA |
| tRNA-Gly-UCC-1-1_ASO | CAACTGCTTGGAAGGCAGCTAT |
| Scrambled ASO | ACGGTCGCCCAGCTGGAAAGGG |

(M fraction) was resuspended in 300 µl HB for protein validation or direct RNA extraction.

RNA was extracted from all three fractions using TRIzol reagent (15596018; Invitrogen) according to the manufacturer's protocol followed by standard isopropanol precipitation, washing, and elution in 20 µl RNase-free water for RT–qPCR analysis.

### Puromycin proximity ligation assay (Puro-PLA)

Puro-PLA experiments were performed as previously described (Tom Dieck et al, 2015) to label newly synthesized GARS1 protein. Briefly, differentiated N2a cells were incubated with 10 µg/ml puromycin for 5 min, washed with warm PBS, and fixed for 10 min in 4% PFA. After fixation and permeabilization, the NaveniFlex Cell (Navinci) protocol was performed using mouse α-GARS1 (1:100) and rabbit α-Puro (1:100) antibodies according to the manufacturer's instructions. Images were acquired using an inverted Zeiss LSM 980 confocal microscope. Spot counting was performed as described below (see the Puro-PLA spot counting section).

### Image acquisition

Fluorescence images were acquired on an inverted Zeiss LSM 980 confocal microscope controlled by ZEN Black software. Images were collected at RT using a 63 × 1.63 numerical aperture (NA) oil-immersion objective and a GaAsP detector. Excitation was provided by 405-, 488-, 594-, and 640-nm laser lines, as applicable to the fluorophores used. Emission was captured in separate channels using preconfigured spectral detection windows in the Zeiss Airyscan module.

For Airyscan imaging (Figs 1A and B, 2D, 3E, G, and I, 4A, B, and E, and 5D), the scan format was variably adjusted to optimally capture individual cells. Images were acquired with a pixel size of 0.01 µm (fulfilling Nyquist sampling criteria where applicable), a line averaging of 2–4, and minimal dwell times to reduce photobleaching. Z-stacks were collected with a step size of 0.10–0.20 µm across a total depth of 2–4 µm. Laser power, detector gain, and offset were optimized to prevent signal saturation and were maintained constant across all conditions and samples within an experiment when quantitative comparisons were performed.

Maximum-intensity projections, or single z-planes where appropriate, were generated using FIJI (ImageJ) for display and quantification. For colocalization quantification in differentiated N2a cells, images were specifically acquired in neurite regions located >20–30 µm from the soma, unless otherwise specified.

### Image analysis

Images acquired by Zeiss LSM 980 with super-resolution (SR) mode were processed by "Airyscan processing" with the "low" option on the ZEN Black software. All subsequent analysis steps were performed in FIJI (Schindelin et al, 2012). A *maximum-intensity projection* (MIP) was applied to all the z-stacks that were acquired. Images were *thresholded* to remove background fluorescence.

### Mask generation

Mask generation was carried out in FIJI (Schindelin et al, 2012) as follows: to delineate the entire cell, a *mask* for the cell body and the neurites was created, using the Hoechst stain and the IF stain (either βIII-Tub, NF, MAP2, GARS1, or RPS3) using the total unthresholded protein channel (Hafner et al, 2019). The starting point of a neurite was manually defined (usually 15–30 µm away from the nucleus), and the neurite outline was traced with the *polyline tool*, then *straightened* and *expanded* by 50–100 pixels. The non-neurite region (cell body) was defined manually by drawing a *polyline* around the soma. Each image was *thresholded*

*to binary* and converted to a mask using FIJI's *convert to a mask* function.

### Fluorescence intensity measurement

Corrected total cell fluorescence (Godwin et al, 2019) for both the cell body and neurites was calculated by subtracting the product of the mean background intensity and the region area from the integrated density in FIJI.

### Spot counting

Spot counting of smiFISH mRNA images in primary neurons was performed using RS-FISH (Bahry et al, 2022) in FIJI (Schindelin et al, 2012). A representative *Gars1* or *Actb* image was used to determine the *anisotropy coefficient* and the settings to discriminate spots (*SigmaDoG* 1.4975, *DoG threshold* 0.01413707, *anisotropy coefficient* 1.4176076650619507, *robust fitting Multiconsensus RANSAC, spot intensity Linear interpolation*). A *multiconsensus RANSAC robust fitting* was used, with *linear interpolation* to identify spots. These parameters were first optimized on a representative image to accurately distinguish true mRNA spots from background fluorescence, then applied consistently across all images in the dataset. Masks delineating the cell body and neurites were generated (see the Mask generation section). A custom pipeline was implemented in Python (available upon request) to detect spots within the area of the soma and neurite masks. Then, the soma and neurite spot densities were calculated by dividing the number of spots within each compartment to their respective area. Finally, the neurite spot density was divided by the soma spot density for Gars1 and Actb mRNAs to calculate the normalized spot density.

### Colocalization analyses

The following steps were performed in FIJI (Schindelin et al, 2012): *Gaussian blur* with a *sigma* of 1 was applied. Then, local contrast enhancement was applied using *Enhance Local Contrast (CLAHE)* (Zuiderveld, 1994) with a *block size* of 127, *max slope* of 6. A threshold was set to remove background by adjusting the *brightness and contrast*. Masks delineating the cell body and neurites were generated (see the Mask generation section). *Analyze particles* was performed to extract the following information for each of the channels: x- and y-coordinates and area. A custom code (available upon request) was implemented in Python to calculate varying percentages of overlap between the two channels inside the neurite mask.

### Puro-PLA spot counting

All steps were performed in FIJI (Schindelin et al, 2012): a threshold was set to remove background by adjusting the *brightness and contrast*. Masks delineating the cell body and neurites were generated (see the Mask generation section). Using the mask, the area of each compartment was calculated using the *Measure* function. The number of puncta was measured using the *Analyze particles* function. Spot density was then calculated by dividing the number of spots in each compartment by its area.

### Bioinformatics analysis

RNA-seq datasets were obtained from their Gene Expression Omnibus (GEO) under accession numbers GSE121069 (Rotem et al, 2017), GSE51572 (Minis et al, 2014), GSE66230 (Briese et al, 2016), and GSE115480 (Middleton et al, 2019); from ArrayExpress under accesion numbers E-MTAB-4978 and E-MTAB-4979 (Zappulo et al, 2017); or directly from published supplementary files (Rotem et al, 2017; Maciel et al, 2018). Normalized counts of Gars1 and Actb mRNAs in neurites and soma (FPKM, TPM, or DEseq2 normalization) were extracted. Neurite-to-soma ratio was then calculated.

## Data Availability

Raw images are deposited in Zenodo (https://doi.org/10.6084/m9.figshare.31449445) and codes used for analysis in GitHub (https://github.com/tylerbrent/Gars1-/tree/main).

## Supplementary Information

## Acknowledgements

We thank Dr. Nitsan Dahan and Dr. Yael Lupu-Haber from the LS&E microscopy unit of the Technion for help and advice, Prof. Eran Perlson from Tel Aviv University for the Actb and Cox7c probes and plasmids, Prof. Shai Berlin and Dr. Ronit Heinrich from the Technion for their support with rat primary neuron culture preparation, Prof. Nabieh Ayoub and his laboratory members from the Technion for the PLA reagents, and members of the Arava laboratory for support. This work was supported by the Israel Science Foundation (748/23 and 1349/23).

### Author Contributions

TB de Leon: conceptualization, formal analysis, investigation, methodology, and writing—original draft, review, and editing.
A Golani-Armon: methodology and project administration.
B Cohen: methodology.
YS Arava: conceptualization, supervision, funding acquisition, and writing—original draft, review, and editing.

### Conflict of Interest Statement

The authors declare that they have no conflict of interest.

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
