## [Reviewer comments · Life Science Alliance]

Locally synthesized Glycyl aminoacyl tRNA synthetase is important for local translation in neurons

Tyler de Leon, Adi Golani-Armon, Bar Cohen, and Yoav Arava

DOI: <https://doi.org/10.26508/lsa.202603630>

Corresponding author(s): Yoav Arava, Technion - Israel Institute of Technology

Review Timeline:	Submission Date:	2026-01-13
	Editorial Decision:	2026-02-20
	Revision Received:	2026-03-05
	Accepted:	2026-03-27

Scientific Editor: Sarita Hebbar

Transaction Report:

February 20, 2026

RE: Life Science Alliance Manuscript #LSA-2026-03630-T

Prof. Yoav Arava
Technion- Israel Institute of Technology
Biology
Israel

Dear Dr. Arava,

Thank you for submitting your manuscript entitled "Locally synthesized Glycyl aminoacyl tRNA synthetase is important for local translation in neurons". Your manuscript was reviewed by three expert reviewers whose comments are appended below. As you will note, the reviewers have found this work on local translation of Glycyl aminoacyl tRNA synthetase interesting and useful. Overall, in line with the reviewers' evaluation, we would be happy to publish your paper in Life Science Alliance pending resolution of the points immediately below and final revisions necessary to meet our formatting guidelines

- We agree with Reviewers 1 and 3 that you must address their concerns on appropriate use of statistical tests, and include a description of sample sizes in terms of biological replicates in the manuscript.
- You must also acknowledge certain limitations regarding the lack of spatio-temporal resolution (all the reviewers and connected to Figures 1, 3, and 5) (2) probe design for isoform specificity (Reviewer 2), and (3) the use of N2a cells (Reviewer 1).
- Finally, we draw your attention to the need to temper claims in the manuscript related to tRNAGly and GARS1 co-localization (Figure 5, Reviewers 2 and 3).

Whilst addressing the above concerns, we request you to ensure that your abstract accurately reflects any claims that are toned down in the manuscript text.

MANUSCRIPT ORGANIZATION AND FORMATTING:

To avoid unnecessary delays in the acceptance and publication of your paper, please read the following information carefully. Full guidelines are available on our Instructions for Authors page, <https://www.life-science-alliance.org/authors>

- In the methods section describing rat primary neuronal culture preparation, kindly confirm that all experiments with animals were performed in accordance with relevant guidelines and regulations. Please also include a statement identifying the institutional and/or licensing committee approving the experiment in accordance with LSA's guidelines (<https://www.life-science-alliance.org/editorial-policies#animals>).
- Please insert a scale bar for Figure S5B, S9. We encourage you to make the scale bars more pronounced for S5B, S8 and S9B and S9C.
- Please add molecular weight markers to the images of blots in Figure 3C and Figure 5B.
- In the methods section, kindly expand on the details for imaging including the microscope used, the details for objectives (Type, name, N.A.), excitation and emission wavelengths/filters and any other pertinent details for description of imaging methods in this work.
- Please upload your main and supplementary figures as single files.
- Please upload your main manuscript text as an editable doc file.
- Please add a Running Title in our system.
- Please add a Summary Blurb/Alternate Abstract in our system.
- Please add Keywords and Category for your manuscript in our system.
- Please add the X and Bluesky handles of your host institute/organization, as well as your own and/or one of the authors, in our system.
- The titles in both the system and the manuscript file must be consistent with each other.
- It is recommended to exclude figures from the manuscript text and upload them separately.
- Please add your main, supplementary figure, and table legends to the main manuscript text after the references section.
- Please upload your Tables in editable .doc or Excel format; They can be included at the bottom of the main manuscript file or sent as separate files.
- Please rename "BIBLIOGRAPHY" to "References."
- Please use the [10 author names et al.] format in your references (i.e., limit the author names to the first 10)
- The "Data Availability" section should be placed after the Materials & Methods section. Please consult our guidelines at

<https://www.life-science-alliance.org/manuscript-prep#format>

- Please add an Author Contributions section to your main manuscript text and the system.
- Please add a Conflict of Interest statement to your main manuscript text.
- Please incorporate the supplementary references into the main references list.
- Please add callouts for Figures S2A-E; S3A-C; S7A-B and Tables 1, 4 to your main manuscript text.
- Please be sure that the authorship listing and order is correct.

LSA encourages authors to provide a 30-60 second video where the study is briefly explained. We will use these videos on social media to promote the published paper and the presenting author (for examples, see <https://docs.google.com/document/d/1-UWCfbE4pGcDdcgzcmiuJI2XMBJnxKYeqRvLLrLSo8s/edit?usp=sharing>). Corresponding or first-authors are welcome to submit the video. Please submit only one video per manuscript. The video can be emailed to contact@life-science-alliance.org

FINAL FILES:

The following items are required for acceptance.

The license to publish form must be signed before your manuscript can be sent to production. A link to the license to publish form will be available to the corresponding author only. Please take a moment to check your funder requirements.

Thank you for your attention to these final processing requirements. Please revise and format the manuscript and upload materials as soon as you are able.

Thank you for this interesting contribution to the literature. We look forward to publishing your paper in Life Science Alliance.

Sincerely,

Sarita Hebbar, PhD
Scientific Editor
Life Science Alliance
<http://www.lsjournal.org>

Reviewer #1 (Comments to the Authors (Required)):

This manuscript describes the localisation of Glycyl tRNA synthetase (Gars1) mRNA to the neurites and its local translation. While the localisation of mRNAs and ribosomes to neurites is studied extensively, information about the local translation of aminoacyl tRNA synthetases is scarce. This manuscript provides useful information about one such critical component of compartmentalised translation in neurons.

1. A few experiments are performed in primary cultured neurons, while others are performed in N2a cells. Since the primary focus is on the compartmentalised presence of Gars1 mRNA and its translation, N2a cells are not an ideal model system. While it might be used for some logistical reasons in some cases, wherever they are used, a proper rationale has to be given.
2. In Fig. 1D and E, a comparison is made only between Gars1 and Actb mRNA and not with tubb3. It is essential to include Tubb3 data in these panels.
3. Fig 2 A, B and C - have only positive control (FMRP), a negative control (such as tubulin) is also required.
4. Fig 3- the chx and puro experiments to show translation-dependent mitochondrial localisation of Gars1 mRNA are quite convincing. But doing this experiment in N2a cells and measuring colocalization at about 10-20 μ M from the cell body defeats the purpose. To show the dendritic/neurite location of this mRNA to mitochondria, using cultured neurons would be more appropriate, where distances beyond 50 μ m from the cell body are generally accepted as a distal dendrite/neurite. In the current form the conclusions are not very convincing.
5. Fig 4 D- No statistics are given for this data; is the difference statistically significant? It is also mentioned n=2-3, which indicates the numbers are not sufficient to apply statistics; this n has to increase, and the data should be presented with statistics.

Reviewer #2 (Comments to the Authors (Required)):

In their work, Arava and co-authors show that glycyl tRNA synthetase mRNA is highly enriched in neurites and is locally translated to produce GARS1 protein. The authors further demonstrate that GARS1 transcripts associate with mitochondria while being translated. They find that the locally synthesized GARS1 protein is closely associated with tRNAGly, and that disruption of this interaction impairs protein synthesis in neurites. Overall, their results highlight some new aspects in the involvement of GARS1 in local translation.

The authors primarily rely on high-resolution confocal microscopy as their main experimental approach. While they include several appropriate controls, this technique remains limited in its spatial resolution. Although I appreciate the thorough and careful methodology employed throughout the study, it is important to acknowledge that certain interactions might be beyond the resolving power of this approach. Therefore, while I am generally supportive of the manuscript, I would suggest some amendments to acknowledge this:

1. P. 4, l. 81-83: the citation of the Ribo-seq at that place of the manuscript appears non-specific. Maybe rephrasing it to indicate that the findings are corroborated by previously published datasets.
2. P. 4, l.90-94: It is unclear how the authors extrapolate from static snapshots to diffusion. Rephrasing this section to clarify that diffusion cannot be directly inferred from the presented microscopy data would prevent overinterpretation.
3. P.7-8: the evidence in this section feels a bit weak, as the probe design does not seem capable differentiating both isoforms that vary by the signal peptide. Would it be possible to distinguish the two isoforms? If not feasible, the authors should rephrase the text to acknowledge that both isoforms cannot be unambiguously identified.
4. P. 11: The evidence in Fig. 5 for tRNAGly and GARS1 co-localization appears somewhat weak. Both tRNAs and GARS1 are flexible molecules, and while it is logical that tRNAGly would interact with its cognate synthetase, the resolution of the microscopy may be insufficient to resolve direct interactions. The observed signal could reflect stochastic movements or transient colocalizations over a larger spatial range than the true enzyme-substrate interaction sphere. It may therefore be more appropriate to move this figure to the supplementary material.

Reviewer #3 (Comments to the Authors (Required)):

De Leon and colleagues describe the results of an interesting set of experiments probing the local translation of an aminoacyl tRNA synthetase (Gars1) and its importance for local translation in neurons. The manuscript is very well written and for the most part, the analysis of the data was properly done (see below for minor points). The number of experimental replicates is appropriate and the conclusions are well supported by the findings.

The question being addressed is very interesting: Does local synthesis of proteins supporting translation play a role in local translation? The experimental approaches performed here to help answer this question were well chosen and designed. I am fairly convinced that the mRNA Gars1 associates with ribosomes locally and are engaged in translation - despite the fact that the authors use one probe to localize ribosomes (RPS3), instead of two, targeting both the small and large subunit of ribosomes,

which would more accurately reflect the presence of a monosome. The lack of temporal resolution (e.g. via live imaging of the association between mRNA and ribosomes) diminishes the impact of the findings, as well as the lack of assessment of dynamic situations, such as depolarization. However, the findings gathered here support the hypothesis that is being tested. The experiments using ASOs in the last figure are very interesting and support the notion that the association of the tRNA synthetase with tRNAs can affect local translation. However, the fact that the control used - a scrambled ASO - induced an effect does create an important confound - does simply the presence of ASOs reduce local protein synthesis? Although my assessment is that the authors did not conclusively demonstrate that the Gars1 mRNA is associated with ribosomes and locally translated in dendrites (using SunTag coupled with live imaging, for example), the authors's examine a very interesting and remarkably unexplored field of research and deserve praise for this.

A minor point: I would recommend the authors re-visit some of the findings that are clearly affected by potential outliers. Examples are Figures 2C, 3D and 4F. I recommend using a statistical approach to determine if there are outliers and remove them if flagged.

Other minor points:

1) Fig S1A: In a separate plot, show the TPM values for Gars1 found in each reference, as a way to show true normalized counts, instead of normalized against some. This would help gain more clarity on the levels of mRNAs localized to dendrites.

2) The puro-PLA is not direct evidence of local translation, as the protein could re-localize from soma to dendrites after translation. The best evidence comes from the super-resolution experiment (Fig 2D), but the use of a single antibody to measure ribosomal localization is still flawed. A better approach would be to incorporate a third antibody against a ribosomal protein belonging to the large subunit and then measure co-localization.

3) Fig 3D : the enrichment of Gars1 mRNA in the mitochondrial fraction is not comparable to the levels of Cox7c mRNA, as stated by the authors. Although the standard deviation bars overlap considerably, the reason they do so is because of a potential outlier in the Gars1 group (reaching delta delta ct of close to 17). I recommend the authors use solid outlier identification methods and re-do this analysis to confirm their findings.

4) Line 159-160: Remove "Strongly" from "[...] strongly support...". It is good evidence but requires additional evidence to strongly support the claims made by the authors.

5) For experiments using CHX and Puromycin, could the authors demonstrate that the total levels of the mRNAs involved in the co-localization are not being affected by the treatment? This would eliminate an important confound of the data interpretation.

6 Statistics: it is unclear why the authors opted for Kruskal-Wallis test, given that the datasets seem to generally have similar variances across groups (as visualized by the standard deviation/ SEM bars). This is a minor issue, but for the sake of rigor I would recommend using One-Way ANOVA with Dunnet's post-hoc.

Please include biological replicates across all figure legends. Also, include the average number of cell sampling per biological replicate.

7) Fig 4E-F: add a representative image of the tRNAlys to the nice dendritic images shown in Fig 4E.

8) Fig 5E and H: Include the comparisons between negative and Scrambled controls, as they may seem to have some difference, particularly in panel H.

RE: Life Science Alliance Manuscript #LSA-2026-03630-T

Prof. Yoav Arava
Technion- Israel Institute of Technology
Biology
Israel

Dear Dr. Arava,

Thank you for submitting your manuscript entitled "Locally synthesized Glycyl aminoacyl tRNA synthetase is important for local translation in neurons". Your manuscript was reviewed by three expert reviewers whose comments are appended below.

As you will note, the reviewers have found this work on local translation of Glycyl aminoacyl tRNA synthetase interesting and useful. Overall, In line with the reviewers' evaluation, we would be happy to publish your paper in Life Science Alliance pending resolution of the points immediately below and final revisions necessary to meet our formatting guidelines

-We agree with Reviewers 1 and 3 that you must address their concerns on appropriate use of statistical tests, and include a description of sample sizes in terms of biological replicates in the manuscript.

Reply: We looked carefully into the statistical tests and added descriptions on sample sizes and biological repeats to each experiment. See our detailed answer in the point-by-point reply below,

-You must also acknowledge certain limitations regarding the lack of spatio-temporal resolution (all the reviewers and connected to Figures 1, 3, and 5) (2) probe design for isoform specificity (Reviewer 2), and (3) the use of N2a cells (Reviewer 1).

Reply:

(1) We were conservative in analyzing our results, especially with regard to spatial resolution. We exclude claims on *distal* neurites in the case of N2a cells or with regard to tRNA-GARS1 interaction (row 201-204). We removed speculative statements regarding diffusion of signals.

(2) We acknowledged that we cannot distinguish between isoforms in the relevant paragraph (p8-9 rows 148-152, 169-172). Of importance, while our GARS1 probes do not distinguish between isoforms, our GFP probes recognize only the transfected GARS1-GFP mRNA, which is exclusively mitochondrial.

(3) we explicitly described the limitations and advantages of using N2a cells in row 213-215.

See our detailed answers in the point-by-point reply below,

-Finally, we draw your attention to the need to temper claims in the manuscript related to tRNAGly and GARS1 co-localization (Figure 5, Reviewers 2 and 3).

Reply: we tempered our claims on their association throughout the manuscript, and revert to a more accurate terminology (proximity). This was done in the Abstract, Results section (page 8) and Discussion.

See our detailed answers in the point-by-point reply below,

Whilst addressing the above concerns, we request you to ensure that your abstract accurately reflects any claims that are toned down in the manuscript text.

Reply: We changed our Abstract to be consistent with these changes.

MANUSCRIPT ORGANIZATION AND FORMATTING:

To avoid unnecessary delays in the acceptance and publication of your paper, please read the following information carefully. Full guidelines are available on our Instructions for Authors page, <https://www.life-science-alliance.org/authors>

-In the methods section describing rat primary neuronal culture preparation, kindly confirm that all experiments with animals were performed in accordance with relevant guidelines and regulations. Please also include a statement identifying the institutional and/or licensing committee approving the experiment in accordance with LSA's guidelines (<https://www.life-science-alliance.org/editorial-policies#animals>).

-Please insert a scale bar for Figure S5, S9. We encourage you to make the scale bars more pronounced for S5B, S8 and S9B and S9C.

Done

-Please add molecular weight markers to the images of blots in Figure 3C and Figure 5B .

Added

-In the methods section, kindly expand on the details for imaging including the microscope used, the details for objectives (Type, name, N.A.), excitation and emission wavelengths/filters and any other pertinent details for description of imaging methods in this work

Done.

-Please upload your main and supplementary figures as single files.

Done

-Please upload your main manuscript text as an editable doc file.

Done

-Please add a Running Title in our system.

Done

-Please add a Summary Blurb/Alternate Abstract in our system.

Done

-Please add Keywords and Category for your manuscript in our system.

Done

-Please add the X and Bluesky handles of your host institute/organization, as well as your own and/or one of the authors, in our system.

Do not have such

-The titles in both the system and the manuscript file must be consistent with each other.

Aligned

-It is recommended to exclude figures from the manuscript text and upload them separately.

Figures were excluded and uploaded separately as TIF

-Please add your main, supplementary figure, and table legends to the main manuscript text after the references section.

Added

-Please upload your Tables in editable .doc or Excel format; They can be included at the bottom of the main manuscript file or sent as separate files.

Added, at the end of the MS

-Please rename "BIBLIOGRAPHY" to "References."

Done

-Please use the [10 author names et al.] format in your references (i.e., limit the author names to the first 10)

Done

-The "Data Availability" section should be placed after the Materials & Methods section. Please consult our guidelines at <https://www.life-science-alliance.org/manuscript-prep#format>

Done

-Please add an Author Contributions section to your main manuscript text and the system.

Added

-Please add a Conflict of Interest statement to your main manuscript text.

Added

-Please incorporate the supplementary references into the main references list.

Done

-Please add callouts for Figures S2A-E; S3A-C; S7A-B and Tables 1, 4 to your main manuscript text.

Callout to Figure S2 appears in row 92, Fig S3 in r95, S7 in r200. Table 1 in r403 and Table 4 in r419

-Please be sure that the authorship listing and order is correct.

The authorship is correct

We uploaded a source code in Code link: <https://github.com/tylerbrent/Gars1-/tree/main>

And images into Zenodo: Images link: 10.6084/m9.figshare.31449445

This is indicated in the Data availability section

No plans

LSA encourages authors to provide a 30-60 second video where the study is briefly explained. We will use these videos on social media to promote the published paper and the presenting author (for examples, see <https://docs.google.com/document/d/1-UWCfbE4pGcDdcgzcmiuJl2XMBJnxKYeqRvLLrLSo8s/edit?usp=sharing>). Corresponding or first-authors are welcome to submit the video. Please submit only one video per manuscript. The video can be emailed to contact@life-science-alliance.org

Unfortunately, we cannot do that.

FINAL FILES:

The following items are required for acceptance.

Uploaded

Uploaded

-- Summary blurb (enter in submission system): A short text summarizing in a single sentence the study (max. 200 characters including spaces). This text is used in conjunction with the titles of papers, hence should be informative and complementary to the title. It should describe the context and significance of the

findings for a general readership; it should be written in the present tense and refer to the work in the third person. Author names should not be mentioned.

Done

The license to publish form must be signed before your manuscript can be sent to production. A link to the license to publish form will be available to the corresponding author only. Please take a moment to check your funder requirements.

Done

OK with that

Thank you for your attention to these final processing requirements. Please revise and format the manuscript and upload materials as soon as you are able.

Thank you for this interesting contribution to the literature. We look forward to publishing your paper in Life Science Alliance.

Sincerely,

Sarita Hebbar, PhD
Scientific Editor
Life Science Alliance
<http://www.lsjournal.org>

Reviewer #1 (Comments to the Authors (Required)):

This manuscript describes the localisation of Glycy tRNA synthetase (Gars1) mRNA to the neurites and its local translation. While the localisation of mRNAs and ribosomes to neurites is studied extensively, information about the local translation of aminoacyl tRNA synthetases is scarce. This manuscript provides useful information about one such critical component of compartmentalised translation in neurons.

1. A few experiments are performed in primary cultured neurons, while others are performed in N2a cells. Since the primary focus is on the compartmentalised presence of Gars1 mRNA and its translation, N2a cells are not an ideal model

system. While it might be used for some logistical reasons in some cases, wherever they are used, a proper rationale has to be given.

Reply: We agree with this comment, and we attempted to perform all key experiments with primary cells and keep the N2a data as supporting, supplementary files. Unfortunately, due to technical limitations, in particular methods that necessitate transfection, we could not accomplish few experimental procedures with primary cells due to their fragility. In such cases we had to revert to N2a cells. We indicate these cases in the text.

2. In Fig. 1D and E, a comparison is made only between Gars1 and Actb mRNA and not with tubb3. It is essential to include Tubb3 data in these panels.

Reply: Tubb3 is an established non-neuritic control mRNA, and was used for that in the imaging panels (Fig1A-C). Because of its low abundance in neurites, distribution analysis is unreliable and we are reluctant to add it to panel D.

For panel 1E (RT-qPCR analysis of fractionated cells), we agree that it is important to include it. However, we have used up all these RNA samples and acquiring new ones will take a significant amount of time. Nevertheless, we routinely used Tubb3 for that purpose and got consistent separation. For example, below are results from an unpublished project in the lab, summarizing six biological repeats, with the same cells and the same fractionation protocol. The depletion of Tubb3 relative to Actb from neurites is clear. We do not want to patch these results to Figure 1E as it will appear awkward.

[Figure removed by editorial staff per authors' request].

3. Fig 2 A, B and C - have only positive control (FMRP), a negative control (such as tubulin) is also required.

Reply: Indeed, such control is important. However, we do not have it for this specific IF, and will not be able to generate such in a timely manner.

Nevertheless, the morphology of the cells (as detected by the TUBB3+Hoechst staining) clearly indicates the separation between soma and neuritic signals.

4. Fig 3- the chx and puro experiments to show translation-dependent mitochondrial localisation of Gars1 mRNA are quite convincing. But doing this experiment in N2a cells and measuring colocalization at about 10-20 μ M from the cell body defeats the purpose. To show the dendritic/neurite location of this mRNA to mitochondria, using cultured neurons would be more appropriate, where distances beyond 50 μ m from the cell body are generally accepted as a distal dendrite/neurite. In the current form the conclusions are not very convincing.

Reply: We note that the zero coordinate of the x-axis in this figure indicates the start point of the analyzed region, and not the cell body. Most of the analyzed regions were usually further away from 10-20 μ M from the cell body. Yet, we did not aim to focus on distal sites, and therefore cannot exclude the possibility that some of the analyzed regions are closer than 50 μ m. Hence, we do not claim that we see co-localization with mitochondria in *distal* regions, but rather that the co-localization is translation-dependent.

5. Fig 4 D- No statistics are given for this data; is the difference statistically significant? It is also mentioned n=2-3, which indicates the numbers are not sufficient to apply statistics; this n has to increase, and the data should be presented with statistics.

Reply: As both types of tRNAs need to be present in neurites to allow local translation, it is not surprising that the difference is not significant. We added NS labels on the bars (using a two-sample t test to account for only 2 repeats for tRNA^{Ala}). We wish to emphasize that this RT-qPCR analysis is aimed to supplement the robust imaging data in panels A-C. Moreover, the critical point of this figure is the different proximity of the tRNAs to Gars1 (Panel F).

Reviewer #2 (Comments to the Authors (Required)):

In their work, Arava and co-authors show that glycyl tRNA synthetase mRNA is highly enriched in neurites and is locally translated to produce GARS1 protein. The authors further demonstrate that GARS1 transcripts associate with mitochondria while being translated. They find that the locally synthesized GARS1 protein is closely associated with tRNA^{Gly}, and that disruption of this interaction impairs protein synthesis in neurites. Overall, their results highlight some new aspects in the involvement of GARS1 in local translation. The authors primarily rely on high-resolution confocal microscopy as their main

experimental approach. While they include several appropriate controls, this technique remains limited in its spatial resolution. Although I appreciate the thorough and careful methodology employed throughout the study, it is important to acknowledge that certain interactions might be beyond the resolving power of this approach. Therefore, while I am generally supportive of the manuscript, I would suggest some amendments to acknowledge this:

1. P. 4, l. 81-83: the citation of the Ribo-seq at that place of the manuscript appears non-specific. Maybe rephrasing it to indicate that the findings are corroborated by previously published datasets.

Reply: This is an excellent suggestion, and we moved the relevant text to line 131.

2. P. 4, l.90-94: It is unclear how the authors extrapolate from static snapshots to diffusion. Rephrasing this section to clarify that diffusion cannot be directly inferred from the presented microscopy data would prevent overinterpretation.

Reply: This is a valid point, and we removed this overinterpretation.

3. P.7-8: the evidence in this section feels a bit weak, as the probe design does not seem capable differentiating both isoforms that vary by the signal peptide. Would it be possible to distinguish the two isoforms? If not feasible, the authors should rephrase the text to acknowledge that both isoforms cannot be unambiguously identified.

Reply: Indeed, our probes cannot distinguish between the two Gars1 isoforms. We clarified this in the text (row 152-155). However, our GFP-tagged construct (Fig. 3I-J) contains an MTS and is therefore exclusively mitochondrial. This mRNA localization data pertained to this variant. We expand on that on row 173-177.

4. P. 11: The evidence in Fig. 5 for tRNAGly and GARS1 co-localization appears somewhat weak. Both tRNAs and GARS1 are flexible molecules, and while it is logical that tRNAGly would interact with its cognate synthetase, the resolution of the microscopy may be insufficient to resolve direct interactions. The observed signal could reflect stochastic movements or transient colocalizations over a larger spatial range than the true enzyme-substrate interaction sphere. It may therefore be more appropriate to move this figure to the supplementary material.

Reply: This is a valid point, and we tempered our statements throughout the MS. Yet, we believe that this is important data for the conclusions of the MS, in

particular when taken together with the impact of the ASO (Fig 5E). We therefore opt to keep this in the main text.

Reviewer #3 (Comments to the Authors (Required)):

De Leon and colleagues describe the results of an interesting set of experiments probing the local translation of an aminoacyl tRNA synthetase (Gars1) and its importance for local translation in neurons. The manuscript is very well written and for the most part, the analysis of the data was properly done (see below for minor points). The number of experimental replicates is appropriate and the conclusions are well supported by the findings.

The question being addressed is very interesting: Does local synthesis of proteins supporting translation play a role in local translation? The experimental approaches performed here to help answer this question were well chosen and designed. I am fairly convinced that the mRNA Gars1 associates with ribosomes locally and are engaged in translation - despite the fact that the authors use one probe to localize ribosomes (RPS3), instead of two, targeting both the small and large subunit of ribosomes, which would more accurately reflect the presence of a monosome. The lack of temporal resolution (e.g. via live imaging of the association between mRNA and ribosomes) diminishes the impact of the findings, as well as the lack of assessment of dynamic situations, such as depolarization. However, the findings gathered here support the hypothesis that is being tested. The experiments using ASOs in the last figure are very interesting and support the notion that the association of the tRNA synthetase with tRNAs can affect local translation. However, the fact that the control used - a scrambled ASO - induced an effect does create an important confound - does simply the presence of ASOs reduce local protein synthesis? Although my assessment is that the authors did not conclusively demonstrate that the Gars1 mRNA is associated with ribosomes and locally translated in dendrites (using SunTag coupled with live imaging, for example), the authors's examine a very interesting and remarkably unexplored field of research and deserve praise for this.

General comments' reply:

A) We include in the revised manuscript imaging data with Rpl10-GFP, to label both ribosomal subunits (see also point #2 below)

B) Regarding the comment on impact of scrambled ASO - its transfection to the cells likely leads to impact on translation, as apparent in Figure 5. Yet, this is exactly the reason why we used it- to separate non-specific impact from the

specific impact of the complementary ASO. The use of scrambled control is a common practice to weed out non-specific effects.

A minor point: I would recommend the authors re-visit some of the findings that are clearly affected by potential outliers. Examples are Figures 2C, 3D and 4F. I recommend using a statistical approach to determine if there are outliers and remove them if flagged.

Reply: We appreciate the reviewer's careful attention to these data points. Following his/her recommendation, we utilized Tukey's fences ($1.5 \times \text{IQR}$) and identified that indeed they have outliers. Nevertheless, removing them from Fig 2C and Fig 4F data and doing a student *t*-test did not change the significance of the observations. Nevertheless, we prefer to maintain all obtained data and use Wilcox test, which takes outliers and deviations into account.

Other minor points:

1) Fig S1A: In a separate plot, show the TPM values for Gars1 found in each reference, as a way to show true normalized counts, instead of normalized against some. This would help gain more clarity on the levels of mRNAs localized to dendrites

Reply: Done. New Figure S1C

2) The puro-PLA is not direct evidence of local translation, as the protein could re-localize from soma to dendrites after translation. The best evidence comes from the super-resolution experiment (Fig 2D), but the use of a single antibody to measure ribosomal localization is still flawed. A better approach would be to incorporate a third antibody against a ribosomal protein belonging to the large subunit and then measure co-localization.

Reply: We added co-localization data with a large subunit protein (Rpl10), tagged with a GFP. This was done in N2a cells and included in new Fig. S4G. Clear co-localization with both RpS3 and RpL10 is observed.

3) Fig 3D: the enrichment of Gars1 mRNA in the mitochondrial fraction is not comparable to the levels of Cox7c mRNA, as stated by the authors. Although the standard deviation bars overlap considerably, the reason they do so is because of a potential outlier in the Gars1 group (reaching delta delta ct of close to 17). I recommend the authors use solid outlier identification methods and re-do this analysis to confirm their findings

Reply: To address this comment, we utilized Tukey's fences ($1.5 \times \text{IQR}$) and confirmed that Replicate 3 represents a statistical outlier for both Gars1 and Cox7c. However, because Cox7c is a rigorously validated positive control in this context, removing this biological replicate disproportionately impacts the statistical power of the control group. Thus, we opted to perform a non-parametric Kruskal-Wallis Test followed by a Dunn's Test.

4) Line 159-160: Remove "Strongly" from "[...] strongly support...". It is good evidence but requires additional evidence to strongly support the claims made by the authors.

Done

5) For experiments using CHX and Puromycin, could the authors demonstrate that the total levels of the mRNAs involved in the co-localization are not being affected by the treatment? This would eliminate an important confound of the data interpretation.

Reply: To address this, we quantified the total Gars1 spots under each treatment (new FigS4F). No significant difference is observed

6 Statistics: it is unclear why the authors opted for Kruskal-Wallis test, given that the datasets seem to generally have similar variances across groups (as visualized by the standard deviation/ SEM bars). This is a minor issue, but for the sake of rigor I would recommend using One-Way ANOVA with Dunnett's post-hoc.

Reply: Shapiro-Wilk and Levene Test were performed to determine if the data meet the appropriate distribution for a One-Way Anova with Dunnett's post-hoc. However, the results led to a non-normal distribution, which prompted us to carry out a non-parametric Kruskal-Wallis test instead.

Please include biological replicates across all figure legends. Also, include the average number of cell sampling per biological replicate.

Reply: These were added to all relevant Figures

7) Fig 4E-F: add a representative image of the tRNALys to the nice dendritic images shown in Fig 4E.

Reply: It may have skipped the Reviewer, but the bottom of panel E includes a representative image of tRNALys

8) Fig 5E and H: Include the comparisons between negative and Scrambled controls, as they may seem to have some difference, particularly in panel H.

Added (both are non-significant (ns)).

March 27, 2026

RE: Life Science Alliance Manuscript #LSA-2026-03630-TR

Prof. Yoav Arava
Technion - Israel Institute of Technology
Biology
Israel

Dear Prof. Arava,

Thank you for submitting your Research Article entitled "Locally synthesized Glycyl aminoacyl tRNA synthetase is important for local translation in neurons", and completing our formatting requests. We also acknowledge your patience whilst we waited for the reviewers' evaluation of your work.

The manuscript was sent back to two of the original reviewers, whose comments are appended below. As you will note, both the reviewers commented that your revised manuscript has addressed most of their previous concerns.

It is a pleasure to let you know that your manuscript is now accepted for publication in Life Science Alliance. Congratulations on this interesting work.

DISTRIBUTION OF MATERIALS:

Again, congratulations on a very nice paper. I hope you found the review process to be constructive and are pleased with how the manuscript was handled editorially. We look forward to future exciting submissions from your lab.

Sincerely,

Sarita Hebbar, PhD
Scientific Editor
Life Science Alliance
<http://www.lsajournal.org>

Reviewer #1 (Comments to the Authors (Required)):

In the revised version, the authors have addressed some of the issues raised in the original review. The issue of using N2a cells for most experiments, which is not suitable for compartmentalised translation, is not satisfactorily addressed. The authors have given technical/logistical difficulties as the reason for this.

Reviewer #3 (Comments to the Authors (Required)):

I am satisfied with the authors' responses to my comments